# SEARCH ARENA: ANALYZING SEARCH-AUGMENTED LLMS

**Mihran Miroyan,**\* **Tsung-Han Wu,**\* **Logan King, Tianle Li, Jiayi Pan, Xinyan Hu**
**Wei-Lin Chiang, Anastasios N. Angelopoulos, Trevor Darrell, Narges Norouzi**
**Joseph E. Gonzalez**
University of California, Berkeley

## ABSTRACT

Search-augmented language models combine web search with Large Language Models (LLMs) to improve response groundedness and freshness. However, analyzing these systems remains challenging: existing datasets are limited in scale and narrow in scope, often constrained to static, single-turn, fact-checking questions. In this work, we introduce **Search Arena**, a crowd-sourced, large-scale, human-preference dataset of over 24,000 paired multi-turn user interactions with search-augmented LLMs. The dataset spans diverse intents and languages, and contains full system traces with around 12,000 human preference votes. Our analysis reveals that user preferences are influenced by the number of citations, even when the cited content does not directly support the attributed claims, uncovering a gap between perceived and actual credibility. Furthermore, user preferences vary across cited sources, revealing that community-driven platforms are generally preferred and static encyclopedic sources are not always appropriate and reliable. To assess performance across different settings, we conduct cross-arena analyses by testing search-augmented LLMs in a general-purpose chat environment and conventional LLMs in search-intensive settings. We find that web search does not degrade and may even improve performance in non-search settings; however, the quality in search settings is significantly affected if solely relying on the model's parametric knowledge. We open-sourced the dataset to support future research.

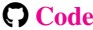 **Search Arena**   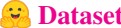 **Code**   🤗 **Dataset**

## 1 INTRODUCTION

Large Language Models (LLMs) have become a popular interface for human–AI interaction, supporting information seeking and task assistance through natural, multi-turn dialogue. However, the capabilities of these models are constrained by their reliance on static training data, which prevents them from effectively handling time-sensitive questions, emerging topics, or niche domains. Search-augmented LLMs aim to bridge this gap by retrieving and using live web data during inference. Access to search enables LLMs to provide up-to-date, domain-specific, and factually verifiable responses (Hilton et al., 2021; Li et al., 2023a). Recent developments (Gemini, 2024; OpenAI, 2024; Perplexity) also reflect the growing interest in search-augmented LLMs.

Despite rapid progress in developing search-augmented LLMs, our understanding of how users interact with these systems—what they ask, how they engage in multi-turn dialogue, and what they expect in return—remains limited. Existing datasets capture interactions with either standalone LLMs (Chiang et al., 2024; Zhao et al., 2024; Zheng et al., 2024) or traditional web search engines (Chen et al., 2024b; Craswell et al., 2020). However, search-augmented LLMs represent a hybrid interface different from both: they not only retrieve information through web search but also rely on their reasoning and conversational capabilities. The most widely used datasets for evaluating these systems (SimpleQA (Wei et al., 2024) and BrowseComp (Wei et al., 2025)) primarily consist of single-turn, monolingual, fact-based queries and are relatively small in scale (typically ≤5k queries). As shown in Figure 1, fact-checking accounts for only one-fifth of real-world user queries; the majority of user prompts, such as seeking analyses, recommendations, or problem-solving guidance, require a

---

[1]\*Equal contribution.

Table 1: **Comparison of Search Datasets.** Unlike prior datasets such as SimpleQA Wei et al. (2024) and BrowseComp Wei et al. (2025), which are static, English-only, single-turn fact-seeking queries, Search Arena evaluates models in diverse, open-ended, multilingual, and multi-turn settings. We release 24,069 conversations with 12,652 preference votes. Further analyses are provided in Section 2.

| Dataset | #Convs | #Langs | Multiturn | Answer/Judge | Conversation Properties | Metadata |
|---|---|---|---|---|---|---|
| SimpleQA | 4,326 | 1 (EN) | No | Short ground truth | Expert-written short factual queries | Verified supporting URLs, topic tags |
| BrowseComp | 1,266 | 1 (EN) | No | Short ground truth | Expert-written challenging prompts with detailed constraints | Topic tags |
| **Search Arena** | 24,069 | 71 | Yes | Human preference | Open-ended, crowd-sourced prompts across diverse intents and topics | Retrieved URLs, full model traces, user intent, and topic tags |

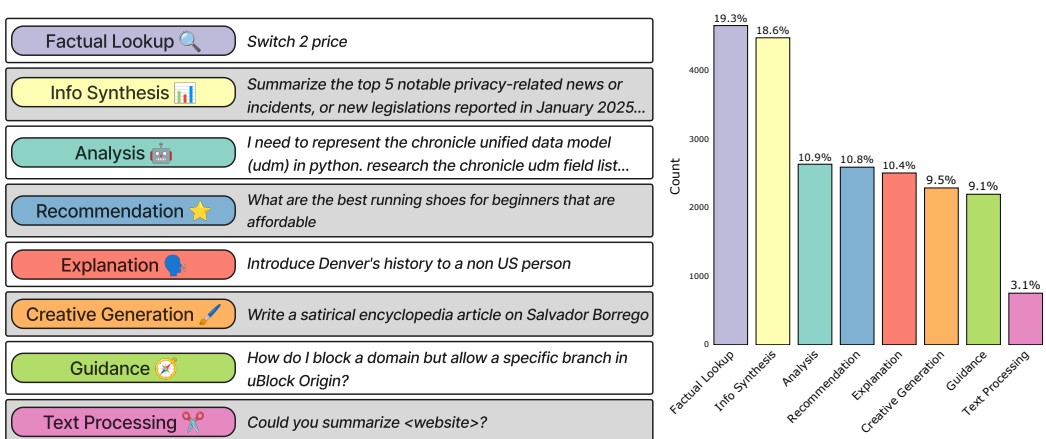

Figure 1: (**Left**) Nine intent categories with representative examples (truncated). In-the-wild prompts are often ambiguous and require real-time web retrieval. (**Right**) Intent distribution across prompts. Most queries require more than a simple fact lookup, ranging from information synthesis to creative generation. The *Other* category is excluded.

combination of factual retrieval, reasoning, and open-ended dialogue. User expectations also extend beyond factual correctness: preferences can be shaped by the number, relevance, and credibility of citations, as well as the presentation style of responses.

To address these gaps, we crowd-sourced the first large-scale human-preference dataset of in-the-wild user interactions with search-augmented LLMs. We developed Search Arena, an open evaluation and data collection platform that presents anonymized side-by-side model outputs in multi-turn settings and collects human votes. During the seven-week deployment period, we gathered and publicly released 24,069 conversations, along with 12,652 paired preference judgments. The dataset spans 11,650 users across 136 countries, 13 models, around 70 languages (including 11% multilingual prompts), and over 5,000 multi-turn interactions. We also introduce a user intent taxonomy in the context of search-enabled human-AI interactions. As detailed in Table 1 and Section 2, Search Arena provides a broad coverage across linguistic and intent features.

We not only analyze user prompts to search-augmented LLMs, but also their preferences. We model user preferences through the Bradley-Terry model (Bradley & Terry, 1952; Chiang et al., 2024; Tianle Li, 2024) and study how different response characteristics interact with user judgments. We find that reasoning, a larger search context window, and longer responses are positively associated with user preferences. Since citations are central to trustworthy web-grounded outputs, we also examine citation features. Our results show that users prefer responses with more cited sources (Figure 4) and those citing tech-related platforms, community blogs, and social networks, but less Wikipedia (Figure 6). While correctly attributed citations positively interact with preferences ($\beta = 0.285$), we observe a positive association with irrelevant citations ($\beta = 0.273$). This raises concerns that users may be overly influenced by citation presence, even when they do not support the associated claims.

We also investigate how search-augmented and non-search models perform across different settings by deploying a non-search LLM in the Search Arena, and a search-augmented LLM in a general-purpose Text Arena. Our results show that conventional LLMs underperform in search-intensive

settings (p-value = 0.009). However, search-augmented models perform comparably in general chat settings, with improved performance on factual lookup queries (p-value = 0.012) and slightly degraded performance on text processing prompts (p-value = 0.077).

In summary, our contributions are as follows: **(1)** we release the first large-scale human-preference dataset of 24k user conversations with search-augmented LLMs, along with 12k preference votes, system metadata, user intents, and prompt topics; **(2)** we present the first analysis of how different characteristics of search-augmented LLMs interact with human preferences; and **(3)** we conduct the first cross-arena evaluation by testing a non-search model in Search Arena and a search-augmented model in Text Arena, reporting that web search augmentation does not hurt and may improve performance across settings, while the internal parametric knowledge of models alone is not sufficient in search-intensive settings.

## 2 HUMAN PREFERENCE DATASET IN SEARCH

We launched Search Arena, an open, crowd-sourced evaluation platform for search-augmented LLMs, on March 18, 2025. The Search Arena is implemented as a separate tab within the Chatbot Arena web application (Chiang et al., 2024), where users interact exclusively with search-enabled models. The search mode interface encourages more search-intensive queries, as users adjust their expectations. During each session, two anonymous models respond to a user query, the user can then cast a vote for their preferred model response. Details on user interface and potential limitations are reported in Appendix A and Section 5.

Between March 18 and May 8, we collected more than 24,000 conversations and 12,000 user votes across 13 models, spanning a range of model configurations (e.g., reasoning models, search context sizes, etc.). The collected dataset contains model identities, the user vote, conversation histories, and system metadata (e.g., reasoning traces, retrieved URLs). Table 1 presents key differences between Search Arena and prior datasets (SimpleQA (Wei et al., 2024) and BrowseComp (Wei et al., 2025)), including dataset scale, prompt characteristics, and available metadata. In the following subsections, we analyze prompt distributions across linguistic and intent dimensions in comparison to existing benchmark datasets.

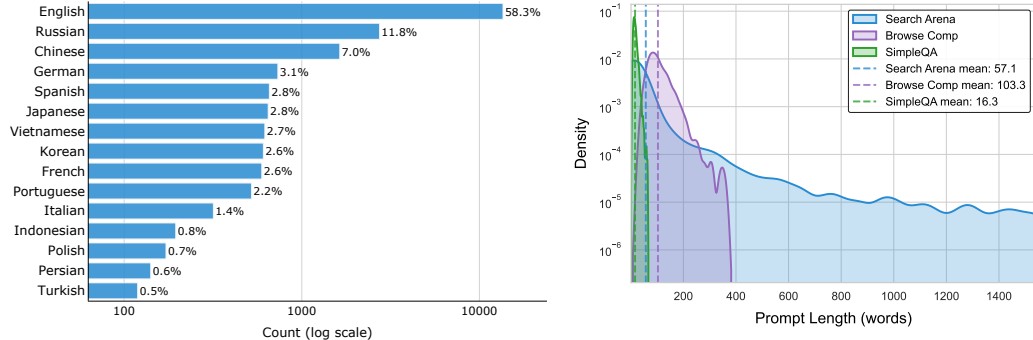

Figure 2: **(Left)** Search Arena prompt language distribution. The dataset is multilingual, spanning over 70 languages, with English prompts accounting for 58.3% of the data. **(Right)** Prompt length distribution of Search Arena (blue), BrowseComp (purple), and SimpleQA (green). Search Arena prompt lengths are more spread out and cover the range of BrowseComp Wei et al. (2025) and SimpleQA Wei et al. (2024) questions.

### 2.1 LINGUISTIC AND CONVERSATIONAL DIVERSITY

Search Arena was collected from 11,650 users across 136 countries, resulting in substantial linguistic and geographic diversity. The prompts span over 70 languages, with 30 represented by at least 10 conversations. English accounts for 58.3% of the data, followed by Russian (11.8%) and Chinese (7.0%). More than 11% of the prompts are multilingual. Figure 2 (Left) shows Search Arena prompt language distribution across the top 15 languages.

Since the platform supports multi-turn interactions, 22.4% of conversations in the dataset are multi-turn, typically clarifications or follow-up queries. Furthermore, as shown in Figure 2 (Right),

SimpleQA prompts are short and fact-oriented (16.3 words on average), while BrowseComp prompts are intentionally constructed to be long and constraint-heavy (103.3 words on average). In contrast, Search Arena includes both brief, under-specified queries as well as longer, detailed requests (57.1 words on average). Additional details and analysis of linguistic features are provided in Appendix B.

## 2.2 INTENT DIVERSITY

Existing search-augmented LLM evaluation datasets focus solely on factuality. To study how in-the-wild user prompts from Search Arena differ from SimpleQA (Wei et al., 2024) and BrowseComp (Wei et al., 2025) questions, we apply an LLM-based dataset differencing framework (Dunlap et al., 2024; Zhong et al., 2022). We also compare Search Arena prompts with the prompt distribution of Text Arena (Chiang et al., 2024), where users' expectations of the models are not influenced by search settings. GPT-4.1 is used for generation and GPT-4.1-mini for ranking of candidate distinguishing properties; more details on the pipeline are provided in Appendix C. The summaries of the top properties reveal high-level pairwise differences between the datasets:

- *Search Arena vs SimpleQA:* Search Arena prompts are broader and more complex, often requiring analysis, synthesis, or creative generation, while SimpleQA prompts are narrowly focused on retrieving specific, factual information with minimal context or interpretation.

- *Search Arena vs BrowseComp:* BrowseComp prompts are structured as investigative challenges, often requiring synthesis of clues under specific constraints, whereas Search Arena prompts prioritize immediate, functional assistance.

- *Search Arena vs Text Arena:* Search Arena prompts focus on real-world factual lookup and decision-making support, often involving up-to-date information. In contrast, Text Arena prompts focus on problem-solving, programming help, and creative generation.

These summaries show high-level differences between prompt distributions across three settings–user prompts to search-augmented LLMs (Search Arena), static factuality questions (SimpleQA and BrowseComp), and prompts to regular LLMs (Text Arena). We also study Search Arena prompts through topic modeling and observe diverse and real-time topics, ranging from market analysis to health discussions (see Figure B.4).

For a structured and in-depth analysis, we introduce a taxonomy of user intent categories. While prior work has explored user intent classification in general dialogue settings (Liu et al., 2024b; Shah et al., 2023), we focus on real-world interactions in search-augmented chat-based settings. The taxonomy includes nine categories: *Factual Lookup*, *Information Synthesis*, *Analysis*, *Recommendation*, *Explanation*, *Creative Generation*, *Guidance*, *Text Processing*, and *Other*. We use secondary labels for ambiguous or multi-purpose prompts. Details on the taxonomy design, as well as descriptions of categories, are provided in Appendix B. We scale the annotation to the full dataset using GPT-4.1, with a manually tuned prompt seeded on 100 examples and validated on 150 multilingual prompts. The resulting Cohen's kappa of 0.812 indicates strong agreement between model- and human-annotated labels. The annotation pipeline and validation details are provided in Appendix B.

Figure 1 shows the resulting intent distribution, along with examples for each category. Factual lookup queries account for only 19.3% of user prompts. The remaining prompts require higher-order capabilities, such as synthesis, guidance, or analysis. We also analyze how linguistic features vary across intents; specifically, we find that factual lookup prompts are typically shorter (17.2 words on average), whereas the remaining set is associated with longer, more complex queries (66.7 words on average). Further analyses of intent categories are provided in Appendix B.

We report comparative analysis with additional related datasets (CORAL, WildChat) in Appendix G.

## 3 PREFERENCE ANALYSES IN SEARCH

With over 12,000 anonymized human preference votes, Search Arena supports fine-grained analysis of how different response features interact with user preferences. We report model performance through both head-to-head win rates as well as scaled coefficients estimated using the Bradley-Terry model (Bradley & Terry, 1952; Chiang et al., 2024) in Appendix D. To analyze how features interact with user preferences, we follow prior work (Tianle Li, 2024) by adding normalized differences between pairwise features to the Bradley-Terry model and reporting the fitted coefficients.

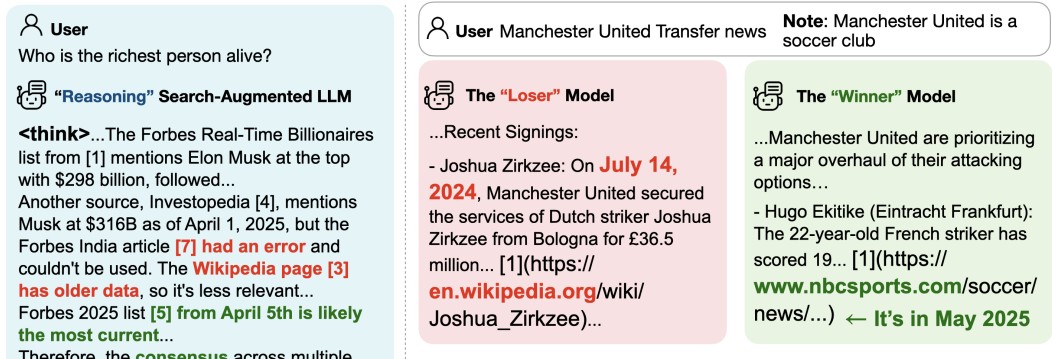

Figure 3: **(Left)** Reasoning trace example, containing multi-document analysis, filtering, and synthesis. **(Right)** Example of a rejected response citing Wikipedia for a sports news question. The preferred response cited the sports division of a news outlet, containing more up-to-date information.

As the correlational analysis in this section relies on user-provided votes, we first clarify their subjective nature and assess their reliability. We do not treat the human-preferred response as the objectively "correct" one, but rather as the response better aligned with the user's personal preferences. In our evaluation and data collection setup, the prompt is tied to the judge (i.e., user): this pairing is critical, as only the original user knows the hidden intent behind their prompt. To verify that users are not voting randomly, we conducted a small study on 100 samples with 3 expert annotators and found expert–user agreement to be 68% when excluding tie votes (random agreement = 50%). Given the inherent subjectivity and the lack of a single "correct" answer for many prompts, some disagreement is expected.

In the following section, our analysis is purely correlational rather than causal, and we view more controlled causal investigation as an important direction for future work. We first focus on two key feature groups: (1) general features such as model type, search depth, and response length (Subsection 3.1), and (2) citation-related features, such as number of citations, citation sources, and citation attributions (Subsection 3.2). We then study how search and non-search models generalize in search-heavy and general chat scenarios in Subsection 3.3. Additional details and further analyses are provided in Appendix D and Appendix E.

## 3.1 GENERAL FEATURES

**Reasoning.** Reasoning models generally perform better in Search Arena prompt distribution, with the top three models achieving over 60.0% average win rates, suggesting that reasoning improves performance in search-augmented chat interactions. Consistent with prior work (Gandhi et al., 2025), we observe prompt analysis, problem decomposition, and backtracking behaviors in reasoning traces. Additionally, we find that reasoning models not only interpret and analyze retrieved content but also rerank sources and filter out irrelevant information. One example is in shown Figure 3 (Left).

**Search Context Size.** Models with high search context windows outperform those with smaller search context. For sonar-pro, the version with high search context has a higher ($p < 0.01$) average win rate (63.9%) compared to the model with medium search context (57.6%). However, the difference is not significant for GPT-4o models with medium and high search context sizes. This finding indicates that models with higher search context retrieve more web sources, leading to more preferred responses; we analyze the effect of citations in Subsection 3.2.

**Response Length.** Consistent with findings from prior work (Chiang et al., 2024; Steyvers et al., 2024; Tianle Li, 2024), we observe that users are biased towards more verbose responses. The Bradley-Terry coefficient corresponding to response length is positive and statistically significant ($\beta_{\text{length}} = 0.334$), indicating that users tend to prefer longer answers. The positive correlation between model score and response length is also shown in Figure 4 (Left). Additionally, Figure 5 (Left) shows the length distribution across eight user intent categories; responses to *Factual Lookup* prompts are much shorter (168.3 words on average) compared to *Creative Generation* (422.8 words) and *Analysis* (393.2 words) prompts. Furthermore, we find that the response length coefficient on *Factual Lookup*

prompts ($\beta_{\text{length, factual}} = 0.156$) is 2.14 times smaller than the effect on the full dataset, suggesting that users prefer less verbose answers to *Factual Lookup* queries compared to other categories.

## 3.2 CITATION FEATURES

Citations are central to the trustworthiness of web-grounded outputs in search-enabled scenarios and a unique feature of Search Arena compared to Text Arena (Chiang et al., 2024). Therefore, we further examine how these factors influence user preference along three dimensions: number of cited sources, types of cited sources, and attribution of inline citations.

**Number of Cited Sources.** We find a positive and statistically significant coefficient for the number of citations ($\beta_{\text{citations}} = 0.209$), indicating that users favor responses with more references. The positive association between model score and response length is shown in Figure 4 (Right). Furthermore, we observe that reasoning models cite fewer sources than non-reasoning models with similar configurations, consistent with our earlier observation that reasoning models filter irrelevant content. Unsurprisingly, models with high search context size end up citing more sources in their final response. Additionally, Figure 5 (Right) shows the distribution of citation counts across prompt intent categories. Notably, responses to *Factual Lookup* prompts contain fewer citations (5.7 on average), compared to *Recommendation* (6.9 citations on average) and *Info Synthesis* (6.8 citations on average) prompts, due to the broader web coverage needed for the latter.

**Types of Cited Sources.** We categorize retrieved URL domains into nine groups (e.g., news, Wikipedia, social media, tech/code platforms, etc.); categorization details appear in Appendix E. Figure 6 (Left) shows source category coefficient estimates with 95% confidence intervals; we also control for citation counts to account for the bias shown earlier. Citing tech-related platforms (e.g., Stack Overflow), community platforms (e.g., Substack), and social media (e.g., TikTok) are positively associated with user preferences with fitted coefficients equal to $\beta_{\text{tech}} = 0.073$, $\beta_{\text{community}} = 0.061$, and $\beta_{\text{social}} = 0.057$, respectively. Surprisingly, citing Wikipedia is negatively correlated with user preferences ($\beta_{\text{wiki}} = -0.071$). To interpret the latter result, we inspect rejected model responses citing Wikipedia and identify two potential explanations: (1) Wikipedia articles are often very lengthy and broad, not directly relevant to a user's question, and (2) citing Wikipedia is not preferred for queries requiring real-time information. A qualitative example appears in Figure 3 (Right).

**Citation Attribution.** We then study how the correctness of citation-to-claim attribution (i.e., whether the inline citation supports the attributed claim) interacts with user preferences. Formally, for each multi-turn interaction, we decompose model responses into a set of claim-citation pairs $(c_i, u_i)$, where $c_i$ denotes a textual claim, and $u_i$ is the corresponding inline citation. For each pair, we evaluate whether the webpage content $D_i$ supports, is irrelevant to, or contradicts the claim $c_i$. This process is automated via an LLM-based pipeline described in Figure 7 through an example. Due to scraping challenges and the high cost of LLM calls, we run the pipeline on roughly 100 conversations per intent category (800 examples in total). The resulting output of each conversation is a set of triplet $\{(c_i, u_i, t_i)\}_{i=1}^{N}$, where $t_i \in \{\text{Support}, \text{Irrelevant}, \text{Contradict}\}$ and $N$ is the total number of claims per conversation. We then compute the number of supporting, irrelevant, and contradicting claims per

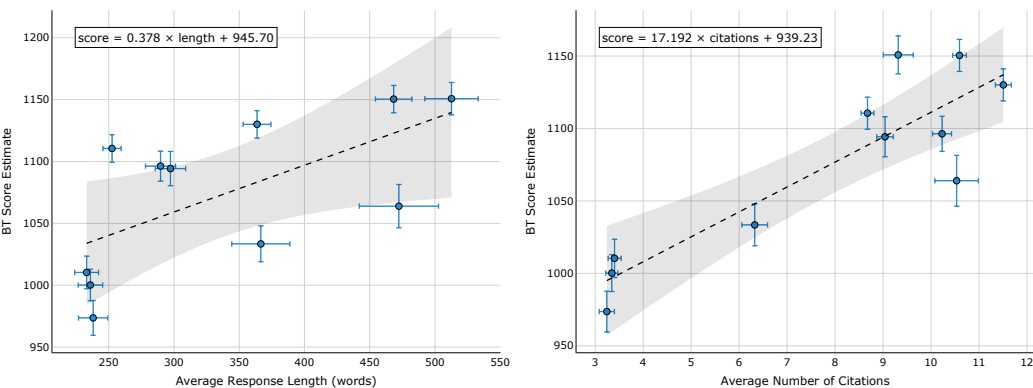

Figure 4: **(Left)** Positive relationship between model score and average response length. **(Right)** Positive relationship between model score and average number of citations.

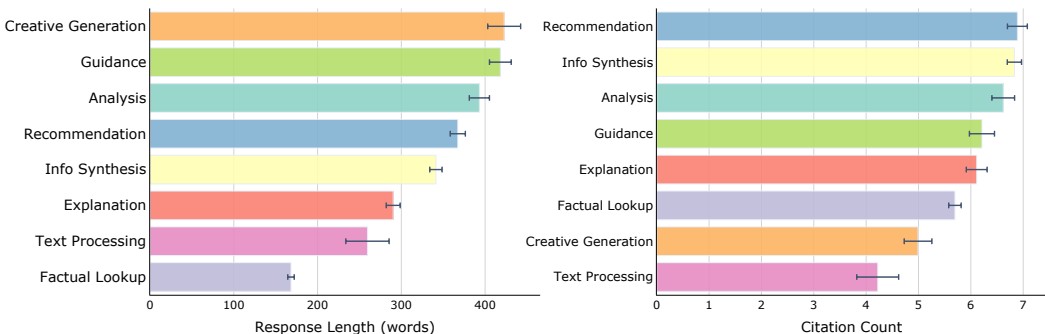

Figure 5: **(Left)** Response length distribution across user intent categories. Responses to *Factual Lookup* prompts are more concise (168.3 words on average) compared to other categories. **(Right)** Citation count distribution across user intent categories. Responses to *Recommendation* (6.9 on average) and *Info Synthesis* (6.8 on average) prompts contain more citations compared to *Factual Lookup* (5.7) and *Text Processing* (4.2) prompts.

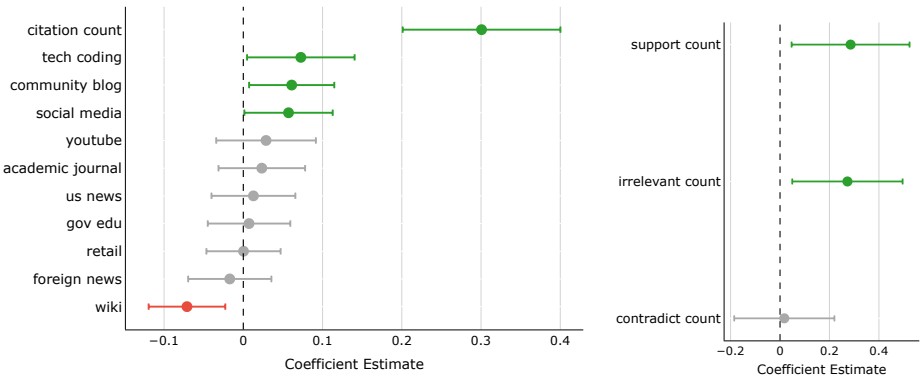

Figure 6: **Citation Control. (Left)** Bootstrapped coefficient estimates of citation features. (1) Users prefer responses with more citations. (2) Citing tech-related and community platforms, as well as social media is positively associated with user preferences. (3) Citing Wikipedia negatively interacts with user preferences. **(Right)** Bootstrapped coefficient estimates of the number of supporting, irrelevant, and contradicting claim-citation pairs. The number of supporting and irrelevant pairs is positively correlated, while the effect of contradicting pairs is not significant.

model response and add them as control covariates in the Bradley-Terry analysis. Implementation details, including scraping tools, parsing logic, and validation process, are provided in Appendix E.

Figure 6 (Right) shows bootstrapped coefficient estimates for the number of supporting, irrelevant, and contradicting claim-citation pairs. While users tend to favor responses with more citations, as shown in Figure 6 (Left), the number of contradicting claim-citation pairs does not show a significant effect on user preference. Furthermore, both supporting ($\beta_{\text{support}} = 0.29$) and irrelevant ($\beta_{\text{irrelevant}} = 0.27$) claims are positively correlated with user preference. Thus, users do not distinguish between supporting and irrelevant citations and generally prefer more citations, even if the citations do not directly support the claims. Upon inspection, in irrelevant citation cases, models may fabricate connections, cite tangentially related sources, or present inferred claims that subtly deviate from the source content. This finding suggests that users may be influenced by the mere presence of citations, rather than their proper attribution to generated claims. We raise this as an open issue for the community: improving citation attribution is critical to ensure that citation-heavy responses are not misperceived as factual and trustworthy.

### 3.3 Cross-Setting Analysis

Search Arena evaluates models where user prompts and expectations are conditioned on models' access to web search. In this section, we study how search and non-search models perform under var-

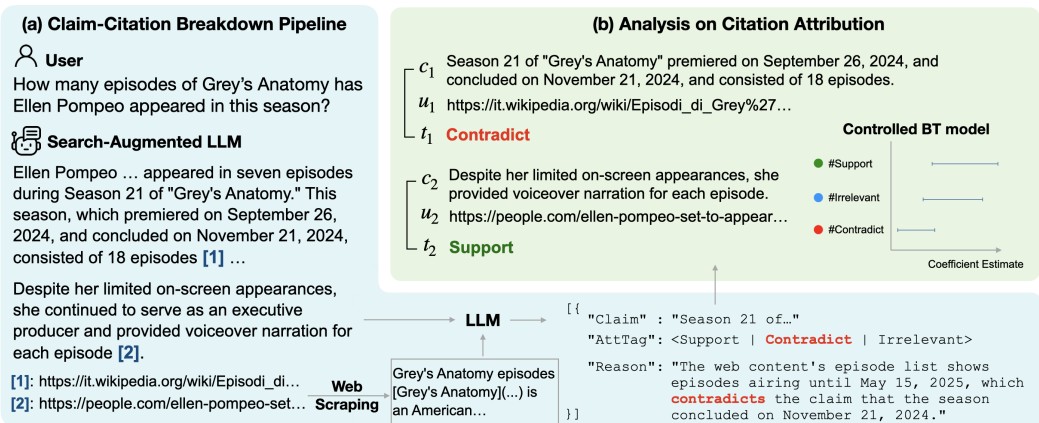

Figure 7: **Experimental Setup for Citation Attribution Analysis.** (a) For each multi-turn conversation, we retrieve the cited web content and use an LLM-based pipeline to decompose each model response into individual claims, followed by citation attribution labeling. (b) For each claim-citation pair $(c_i, u_i, t_i)$, we compute turn-level citation counts across the three categories and add corresponding features to the Bradley-Terry model. Results are shown in Figure 6 (Right).

ious prompt distributions and user expectations–specifically, Search Arena vs Text Arena. Additional results on model performance across benchmarks are provided in Appendix F.

To investigate performance differences across settings, we deployed Gemini-2.5 Pro Experimental (Gemini, 2025)–with and without access to web search–to the Search and Text Arenas. In the Text Arena, the search model only competed against its non-search version; in the Search Arena, the non-search model competed against all other supported search models. Additionally, inline citations were disabled for the search model in the Text Arena to avoid vote bias. In the Text Arena, users typically assume that models operate in closed-book settings without access to external information, while in the Search Arena, user expectations are explicitly conditioned on the search setting. This setup enables us to examine whether and under what conditions access to web search enhances or degrades model performance.

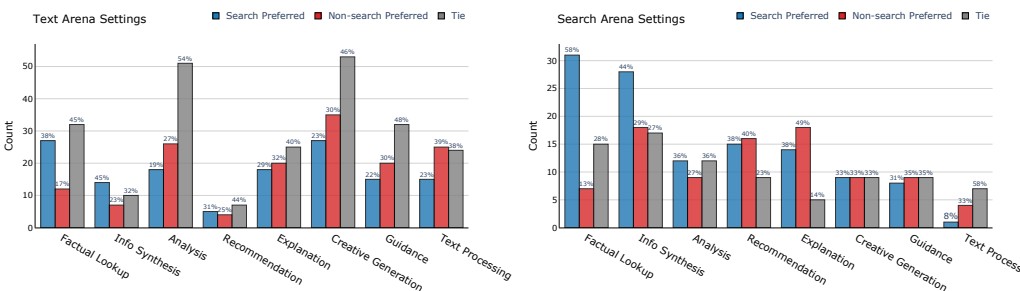

Figure 8: **Cross-Arena Vote Distribution** across Text Arena **(Left)** and Search Arena **(Right)** broken down into user intent categories. Users prefer the search model for *Factual Lookup* and *Info Synthesis* queries in both settings and the non-search model for *Text Processing* queries in Text Arena.

**Text Arena Setting.** We collected 544 battles between the search and non-search versions of the model, yielding 245 ties (45%), 143 search-preferred (26%), and 156 non-search preferred votes (28%). We observe a high proportion of tie votes, and the difference between search-preferred and non-search-preferred votes is not statistically significant (p-value = 0.244). On aggregate, search and non-search models have comparable performance in the Text Arena. To further analyze performance differences by prompt type, we apply the intent classification pipeline from Subsection 2.2 on the 544 collected Text Arena prompts. The distribution of votes by intent class is shown in Figure 8 (Left). We observe a high proportion of ties in *Analysis*, *Creative Generation*, and *Guidance* queries, indicating that there are no significant differences between model responses for these types of prompts. However, for *Factual Lookup* (p-value = 0.012) and *Info Synthesis* (p-value = 0.095), the difference

is more pronounced in favor of the search model. Thus, the search-augmented model is preferred for knowledge acquisition tasks–even in the absence of well-defined user expectations–because web-grounded responses typically provide precise data, statistics, dates, names, and domain-specific terminology. We also note that the difference in performance for *Text Processing* (p-value $= 0.077$) queries favors the non-search model. In these cases, the non-search model often provides structured responses (e.g., numbered or bulleted lists, headings), which suggests that presentation style may impact user evaluations.

**Search Arena Setting.** We collected 315 pairwise battles between a non-search model and search-augmented models in Search Arena, with 99 ties (31%), 126 search-preferred (40%), and 90 non-search-preferred votes (29%). The difference between search and non-search-preferred votes is statistically significant (p-value $= 0.009$); the non-search model underperforms under a search-conditioned distribution. A detailed vote distribution by user intent is shown in Figure 8 (Right). Compared with the Text Arena (Figure 8 (Left)), tie votes are less frequent across all categories, indicating that the differences between model responses are more pronounced. The difference is most expressed for *Factual Lookup* (p-value $= 5.8 \times 10^{-5}$) and *Info Synthesis* (p-value $= 0.092$) queries.

These cross-arena experiments demonstrate that search augmentation does not hurt performance in non-search settings and can improve responses to queries related to information retrieval and synthesis. However, removing web search significantly hurts model performance in search settings.

## 4 RELATED WORK

**Large Language Models.** LLMs have made impressive advances in language understanding, dialogue generation, and reasoning, enabled by techniques such as large-scale pretraining (Anthropic, 2024; Bai et al., 2023; Brown et al., 2020; Grattafiori et al., 2024; Liu et al., 2024a; Touvron et al., 2023), chain-of-thought prompting (Kojima et al., 2023; Wei et al., 2023; Shinn et al., 2023; Wang et al., 2023; Yao et al., 2023a), and reinforcement learning with human feedback (RLHF) (Bai et al., 2022; Ouyang et al., 2022a). The dataset and evaluation landscape has moved from static benchmarks (Hendrycks et al., 2021a; Joshi et al., 2017; Hendrycks et al., 2021b) toward more challenging settings such as deep reasoning (Li et al., 2024; Kazemi et al., 2025; Zeng et al., 2024; Lin et al., 2025), coding (Jimenez et al., 2024; Jain et al., 2024), and open-ended dialogue under crowd-sourced evaluation (Chiang et al., 2024; Zhao et al., 2024). While crowd-sourced setups can mitigate concerns over data contamination in pretraining (Yang et al., 2023; Cheng et al., 2025b), recent work shows potential biases and oversights in human preferences (Clark et al., 2021; Wu & Aji, 2025).

As LLMs gain tool-use capabilities (e.g., APIs, code interpreters, and web browsers) (Hilton et al., 2021; Schick et al., 2023; Grattafiori et al., 2024; Patil et al., 2024; Yao et al., 2023b), domain-specific benchmarks and datasets have emerged to analyze and evaluate LLMs under different environments (Zhou et al., 2024; Li et al., 2023b; Liu et al., 2023). In the context of web search, several search-augmented LLMs have been developed (OpenAI, 2024; Perplexity; Gemini, 2024), which retrieve live information to support better reasoning. However, existing benchmarks such as SimpleQA (Wei et al., 2024) and BrowseComp (Wei et al., 2025) are limited to single-turn, fact-based, monolingual queries. Although WebArena (Zhou et al., 2024) contains diverse user prompts and a web-based interface, it emphasizes closed-world web navigation tasks rather than open-ended search, reasoning, and dialogue. We introduce Search-Arena, the first large-scale, crowd-sourced dataset for search-augmented LLMs with human preference signals, covering diverse intents, topics, and multi-turn interactions across 70+ languages, collected through a transparent, open platform.

**Traditional and LLM-Integrated Information Retrieval (IR).** Information retrieval is a long-standing task, with early methods like BM25 (Robertson et al., 1994), PageRank (Page et al., 1998), and embedding-based approaches (Monir et al., 2024; Huang et al., 2020). Several static benchmarks (Thakur et al., 2021; Muennighoff et al., 2023) and large-scale web search datasets with user logs and preference signals (Voorhees & Harman, 2005; Craswell et al., 2020; Chen et al., 2024b) have enabled robust evaluation of retrieval systems and comprehensive studies on user behavior.

With the rise of LLMs, information retrieval has moved beyond traditional search and become integrated into LLM workflows. In Retrieval Augmented Generation (RAG) settings, retrieved text is appended to the input prompt (Lewis et al., 2021; Guu et al., 2020; Hilton et al., 2021; Chuang et al., 2025); Datasets for evaluating these LLM-IR systems span needle-in-a-haystack retrieval (Kamradt, 2023; Wu et al., 2025), citation attribution (Zhang et al., 2024; Abolghasemi et al., 2024), and general question answering (Yang et al., 2018; Han et al., 2024; Wei et al., 2024; 2025). More

recently, this line of work has been extended from direct single-round information-seeking to natural, conversational search scenarios (Gao et al., 2023; Mo et al., 2025; Cheng et al., 2025a; Adlakha et al., 2022), and then to search-augmented LLM systems operating over the open web (Hilton et al., 2021; Schick et al., 2023; Yao et al., 2023b). We view search-augmented LLM systems as a variant of RAG, with clear differences, including (1) "unbounded" search space over the whole open web and (2) multi-round agent-based query refinement, search, and generation. However, for search augmented LLMs, existing benchmarks lack full human-AI interaction traces, including queries, retrieved documents, responses, and preferences, which are needed for human-centric analysis as done in traditional IR (Craswell et al., 2020; Chen et al., 2024b). To fill this gap, we introduce Search Arena, a large-scale open dataset that enables human-centered analysis of this emerging interface.

## 5 LIMITATIONS, CONCLUSION, AND FUTURE WORK

**Limitations, Future Work, and Broader Impact.** Crowd-sourced data analysis provides valuable insight into real-world user preferences but comes with limitations and broader social implications. First, the collected data may reflect demographic skews and may not be fully representative of the broader population, as not all users choose to vote and human judgments are inherently subjective (Chen et al., 2024a; Clark et al., 2021). Second, because conversational data with human preferences is personal and potentially valuable for model improvement (Ouyang et al., 2022b), its release requires careful consideration of both privacy and equitable access across the community.

To address these concerns, we anonymized model responses and randomized their left/right placement to reduce known human biases. During the data collection period, no early access to data was granted to model providers, nor were any pre-release models deployed on the platform. We obtained user consent at interaction time and enforced a strict privacy policy. To aid responsible interpretation, we also analyzed known biases (e.g., response length; Subsection 3.1) and reported user demographics in Figure B.1 and Figure B.5. Additional details are provided in Appendix A.

We identify several limitations in our current study that open avenues for future research. While our primary focus is dataset construction and interaction analysis, the Search Arena dataset facilitates the following directions:

- **Objective metrics:** As detailed in Section 3, user preferences are often influenced by surface-level features, such as citation counts or source reputation, rather than the intrinsic quality of the cited sources and the generated content. Consequently, user-provided labels may diverge from objective standards like factuality, citation-claim consistency, and source specificity. Future work should expand the evaluation framework to include these objective metrics and analyze their correlation with human preference signals to better understand the gap between perceived credibility and actual groundedness.

- **Offline evaluation:** Our current evaluation relies on an online human-in-the-loop setup, which, while robust, restricts the scalability of testing new models. Future work can leverage our dataset to construct automated benchmarks for offline evaluation. This could involve training reward models or LLMs that approximate human preferences, potentially combining them with the objective metrics discussed above to create a comprehensive scoring rubric.

- **Model development:** Our cross-arena and preference analyses reveal that blindly augmenting models with search is not always optimal. Future research can utilize these insights to improve both the search module (e.g., better source filtering) and the backbone LLM (e.g., optimizing policies for *when* to trigger search versus relying on parametric knowledge).

**Conclusion.** We present Search Arena, the first large-scale dataset and analysis of human interactions with search-augmented LLMs. Spanning over 24k multi-turn conversations and 12k human preference votes across more than 70 languages, the dataset captures a broader range of user intents and topics than prior benchmarks. Our analysis reveals that user preference is positively associated with citation count and certain source types; however, models do not always cite correctly, highlighting a key challenge in trustworthy systems. The cross-arena experiment further shows that search-augmented and non-search models behave differently across settings. We release the full dataset to support future research in search-augmented LLMs and human-centric analyses.

ACKNOWLEDGMENTS

We are deeply grateful to Lisa Dunlap for her invaluable feedback and thoughtful discussions. Sky Computing Lab is supported by gifts from Accenture, AMD, Anyscale, Cisco, Google, IBM, Intel, Intesa Sanpaolo, Lambda, Lightspeed, Mibura, Microsoft, NVIDIA, Samsung SDS, and SAP. Authors, as part of their affiliation with UC Berkeley, were supported in part by the National Science Foundation, US Department of Defense, and/or the Berkeley Artificial Intelligence Research (BAIR) industrial alliance program, as well as gifts from Amazon.

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

APPENDIX

## A  SEARCH ARENA PLATFORM

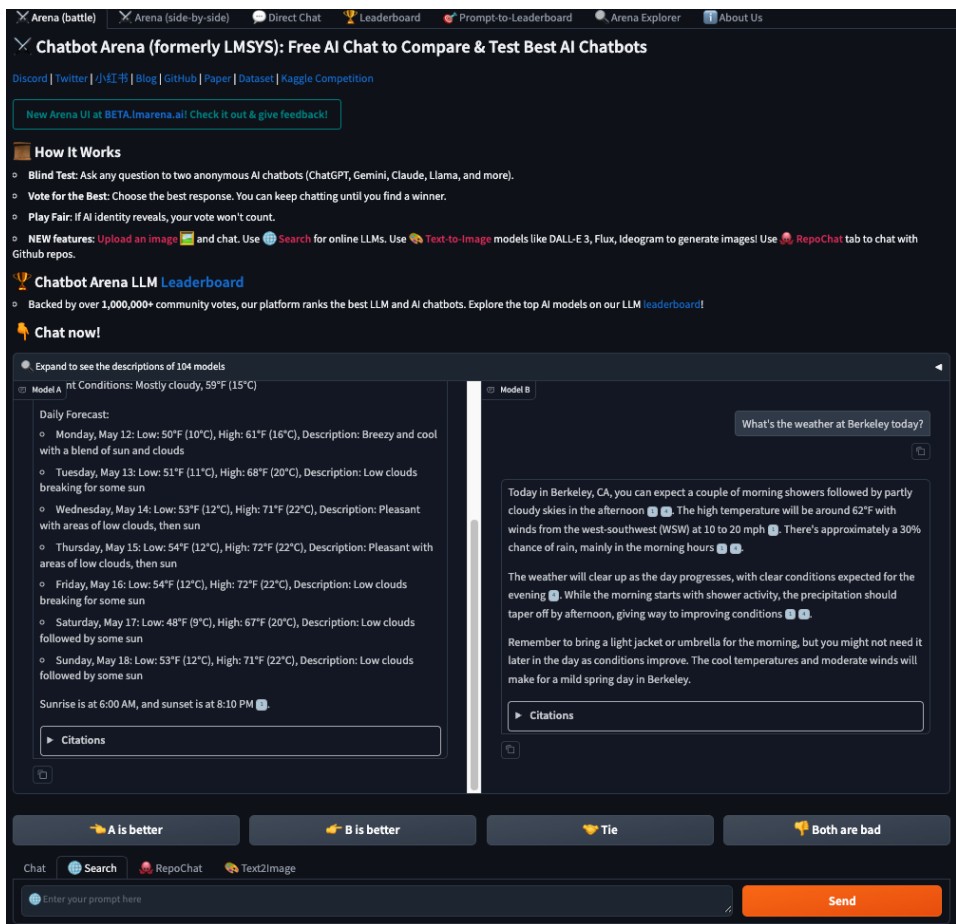

Figure A.1: **Search Arena User Interface.** Search Arena is integrated into the Chatbot Arena ecosystem, sharing the same front-end design and entry point. As shown at the bottom, it is implemented as a separate tab. The central panel displays a side-by-side chat interface with two anonymous models, each providing responses that include clickable inline citations and expandable reference links. Users can engage in multi-turn conversations and cast a vote at any time during the conversation using the four feedback options (A is better, B is better, Tie, and Both are bad).

As described in Section 2, Search Arena (https://arena.ai/search) is an open, crowd-sourced evaluation platform for search-augmented LLMs, launched on March 18, 2025. It should be noted that the platform's user interface has been changed and the data for this study has been collected through the legacy interface (see Figure A.1). This section outlines our data collection and release protocols, followed by a description of the supported models and key design decisions.

## A.1 DATA

**Data Collection.** The Search Arena platform does not require user login, but users must explicitly accept the Terms of Service before using the platform. Once accepted, users access a dedicated tab within Chatbot Arena to interact with search-enabled models. This setup naturally leads to more information-seeking queries compared to the default Text Arena interface. For each prompt, responses from two anonymous models are displayed side-by-side, and users may cast a preference vote at any time during the interaction. The user interface is shown in Figure A.1. The full text of Terms of Service is at Figure A.2.

Over the course of a 7-week data collection period, the platform recorded an average of roughly 800-1,500 conversations per day. After filtering out examples with server errors, inconsistent configurations, or other quality issues, we retain approximately 24,000 conversations, about half of which include human preference votes. Figure A.3 shows daily traffic trends, with spikes aligned to major model updates or platform announcements, and dips on weekends.

---

**Terms of Service**

Users are required to agree to the following terms before using the service: The service is a research preview. It only provides limited safety measures and may generate offensive content. It must not be used for any illegal, harmful, violent, racist, or sexual purposes. Please do not upload any private information. The service collects user dialogue data, including both text and images, and reserves the right to distribute it under a Creative Commons Attribution (CC-BY) or a similar license. You may only use this website for your personal or internal business purposes. You must not access the website programmatically, scrape or extract data, manipulate any leaderboard or ranking, or authorize or pay others to access or use the website on your behalf. Unauthorized use may result in suspension or termination of your access, including access by your organization.

---

Figure A.2: Term of Service of the Search Arena Platform.

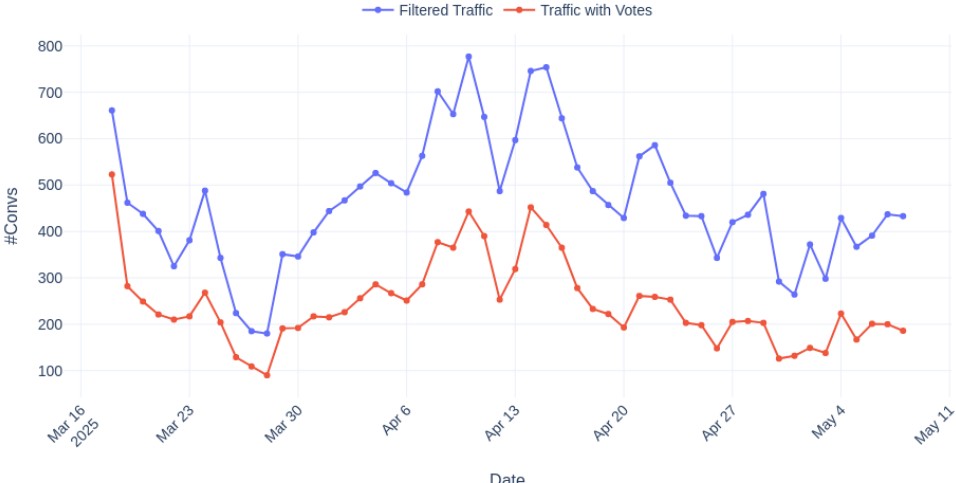

Figure A.3: Daily traffic on the Search Arena platform.

**Data Release Policy.** Per our Terms of Service (Figure A.2), all conversation data is released under a Creative Commons Attribution (CC-BY) license. To protect user privacy, we apply automated de-identification using Google's Data Loss Prevention (DLP) API, which removes personal identifiers such as email addresses, credit card numbers, and API keys. Approximately 2% of examples were flagged by this process. Users may also contact the platform provider to request their records be removed from the released dataset through privacy@arena.ai. Details on the privacy policy of the platform are provided here.

Table A.1: List of supported models and their configurations on Search Arena. [†] We evaluate OpenAI's web search API, which differs from the search feature in the ChatGPT product.

| Provider | Model | Base Model | Details |
|---|---|---|---|
| Perplexity | pplx-sonar | sonar | Default config |
| | pplx-sonar-pro | sonar-pro | Default config |
| | pplx-sonar-pro-high | sonar-pro | `search_context_size=high` |
| | pplx-sonar-reasoning | sonar-reasoning | Default config |
| | pplx-sonar-reasoning-pro-high | sonar-reasoning-pro | `search_context_size=high` |
| Gemini | gemini-2.0-flash-grounding | gemini-2.0-flash | With Google Search enabled |
| | gemini-2.5-flash-grounding | gemini-2.5-flash | With Google Search enabled |
| | gemini-2.5-pro-grounding | gemini-2.5-pro-exp-03-25 | With Google Search enabled |
| OpenAI[†] | api-gpt-4o-mini-search-preview | gpt-4o-mini | Default config |
| | api-gpt-4o-search-preview | gpt-4o | Default config |
| | api-gpt-4o-search-preview-high | gpt-4o | `search_context_size=high` |
| | api-gpt-4o-search-preview-high-loc | gpt-4o | `user_location` feature enabled |

## A.2 MODELS

Search Arena currently supports 12 search-augmented LLMs from Perplexity, Gemini, and OpenAI, as summarized in Table A.1. Unless otherwise noted, we group models with different inline citation styles (elaborated below) of the same base model. Each model is accessed via its provider's public API using the default configuration, including `search_context_size=medium` for Perplexity and OpenAI models. We also evaluate several additional variants, including (1) OpenAI's `gpt-4o-search-preview` with `user_location` enabled and (2) Perplexity and OpenAI models configured with `search_context_size=high`. Figure D.1 shows the number of battles per model.

**Model Anonymity and Citation Style Control.** Each provider uses a distinct citation style, which may unintentionally reveal model identity. At the same time, citation formatting can influence user preferences. To manage this tradeoff, we implement a citation *style randomization* mechanism: model outputs are rendered using either a standardized format or the provider's original style. This approach reduces the risk of user-side de-anonymization while allowing us to study how citation style affects user behavior. We found that the inline citation style does not significantly affect user preferences and model rankings.

## B DATA STATISTICS

### B.1 LINGUISTIC FEATURES

Users' demographic information in Search Arena, based on country codes extracted from IP addresses, is shown in Figure B.1. The dataset includes 11,650 unique users across 136 countries, with the United States (18.7%), Russia (9.9%), Germany (5.9%), and China (5.5%) as the top four.

Language distribution (of the top 50 languages) of Search Arena prompts is shown in Figure B.5. The prompts span 71 languages with English (56.4%), Russian (11.4%), and Chinese (6.8%) as the top three.

As shown in Figure B.2, Search Arena prompt lengths vary across intent categories. As expected, *Text Processing* (85.1 words) and *Analysis* (72.7 words) prompts are generally longer, while *Explanation* (24.7 words) and *Factual Lookup* (16.3 words) prompts are shorter.

Search Arena conversations are multi-turn, with 22.4% of all conversations containing more than one turn. Specifically, there are 3,288 conversations with 2 turns, 966 with 3 turns, and 460 with 4 turns. The distribution is shown in Figure B.3.

### B.2 USER INTENT ANNOTATION

In Subsection 2.2 and Figure 1, we briefly introduced the intent annotation pipeline and its high-level findings. Here, we provide additional details of the pipeline.

First, the three co-authors of the paper did open-text annotations on 100 English prompts randomly sampled from the collected dataset. The annotators then met to consolidate their annotations into a taxonomy of primary and secondary user intent categories, including definitions and representative

examples. The final taxonomy includes the following nine intent categories: *Factual Lookup*, *Information Synthesis*, *Analysis*, *Recommendation*, *Explanation*, *Creative Generation*, *Guidance*, *Text Processing*, and *Other*. Full category descriptions are shown in Table B.1.

To validate the taxonomy, the annotators labeled a subset of 100 prompts. The human inter-annotator agreement, measured by Cohen's Kappa, ranged from 0.65 for primary labels to 0.79 when considering top-two matches (substantial agreement).

We then scaled the annotation to the full dataset using GPT-4.1 (OpenAI, 2025), using the same in-context examples from Table B.1. The total annotation cost was approximately \$20 USD. To evaluate label quality, we compared GPT-4.1 annotations with human labels on 150 samples drawn from the top three languages in our dataset (English, Russian, and Chinese). The resulting Cohen's Kappa score of 0.812 on top-2 intents indicates strong agreement. The full GPT-4.1 prompt is provided below.

---

💬 **Prompts for LLM-based Intent Classification**

```
You are an impartial classifier.

TASK 1:  Primary intent:  choose one category that best matches the
user's intent, always ask yourself, what is the user trying to get
the model to do with this query.  Considering them following the
categories one by one in order.
TASK 2:  If there is a clear secondary goal, choose one additional
intent label that is also present in the query.  Otherwise, return
"Unassigned" if no clear second goal exists.

Allowed intent categories:
    1. Text Processing

    2. Creative Generation

    3. Factual Lookup

    4. Info Synthesis

    5. Analysis

    6. Recommendation

    7. Explanation

    8. Guidance

    9. Other
Here are a few examples that you should follow, try to generalize
beyond these examples but hold true to their semantic meaning:

{FEW SHOT IN-CONTEXT EXAMPLES}

Respond only in valid JSON, only choosing exactly one category to fill
the values:

{
    "primary_intent": "<primary intent label>",
    "secondary_intent": "<secondary intent label>",
    "reasoning": "<your reasoning for the classification>",
}

User query:  {USER PROMPT}
```

---

Table B.1: User intent taxonomy including category descriptions and examples.

| Category | Description | Examples |
|---|---|---|
| **Text Processing** | Users request linguistic tasks such as summarizing, translating, paraphrasing, or refining text. | • Summarize *<website>*.
• Translate this paragraph from Spanish to English.
• Rephrase this email professionally for an investment-banking application. |
| **Creative Generation** | Users request original creative outputs, including fictional scenarios, satire, poetry, storytelling, or other forms of artistic language generation. | • Compose a witty review of ITV's *Big Brother*.
• Create a humorous poem about *SpongeBob*.
• Write a short fantasy story set on Mars. |
| **Factual Lookup** | Users seek to retrieve a precise, objective fact or specific information (what/who/when-like questions). | • Who invented lambda calculus?
• How many planets are in our Solar System?
• When was the Mona Lisa painted? |
| **Info. Synthesis** | Users seek a concise, aggregated summary that combines facts or perspectives from multiple sources. The focus is on integrating factual content without requiring reasoning, subjective interpretation, or personalization. | • List the key functions of the U.S. Congress.
• Summarize the main events of the French Revolution. |
| **Analysis** | Users seek a reasoned judgment or breakdown of a topic, often involving comparison, evaluation, weighing different perspectives, and syntheszing material from many sources. Often open-ended. | • Analyze the pros and cons of nuclear energy.
• When will we reach artificial general intelligence?
• What were the strategic implications of the Cuban Missile Crisis? |
| **Recommendation** | Users request suggestions or advice tailored to particular constraints, preferences, or specified criteria. Often implies personalization or ranking. | • Best laptop for machine learning under $1500.
• I like literary fiction—what books should I read?
• What programming language should I learn for web development? |
| **Explanation** | Users seek detailed clarifications, educational insights, or thorough elaborations aimed at better understanding a concept, process, or phenomeneon. May ask "why" or "how" something works without implying action. | • What led to the fall of the Roman Empire?
• Can you explain the Heisenberg uncertainty principle?
• How does gradient-descent optimization work in machine learning? |
| **Guidance** | Users request instructions, procedures, or practical advice intended to accomplish specific tasks, typically involving sequential steps or troubleshooting. Usually framed as "how to" or involving action. | • How do I install Docker on Ubuntu?
• How do I renew my passport online?
• How do I replace the battery in a MacBook Air? |

| Category | Description | Examples |
|----------|-------------|----------|
| **Other** | Prompts that don't fit neatly into any of the categories above. Prompts may be malformed, declarative in nature, or lacking in meaningful user intent. | • pizza.
• The table hit my head.
• Talk about something please. |

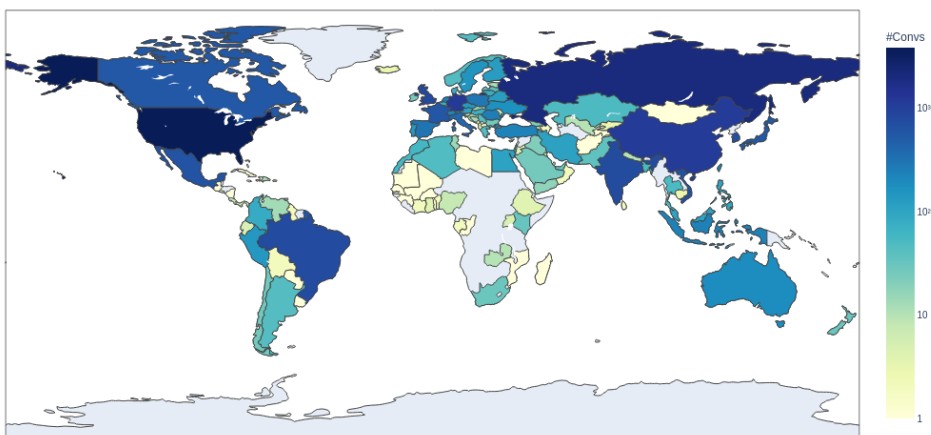

Figure B.1: **Search Arena Users' Demographics.** Search Arena data includes 11,650 unique users across 136 countries.

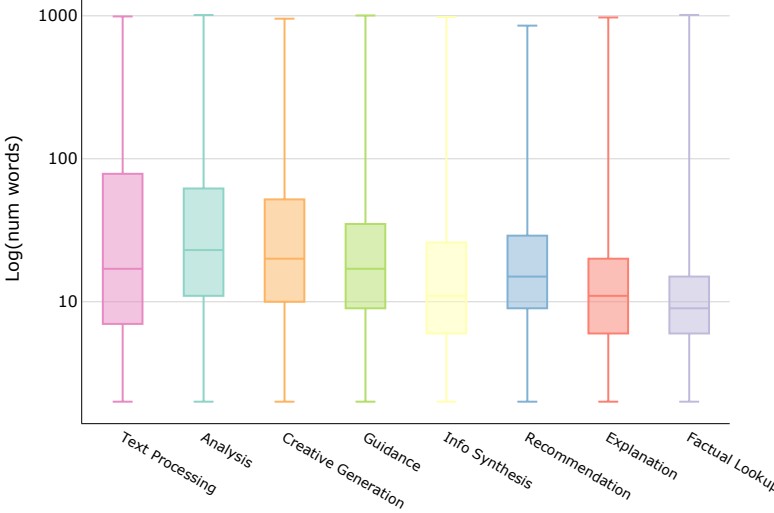

Figure B.2: **Search Arena Prompt Length Distribution by Intent.** Prompt lengths vary across intent categories. *Text Processing* (85.1 words) and *Analysis* (72.7 words) prompts are generally longer, while *Explanation* (24.7 words) and *Factual Lookup* (16.3 words) prompts are shorter.

### B.3 TOPIC MODELING

To analyze topic diversity across benchmarks, we apply the BERTopic framework (Grootendorst, 2022) to the 11,764 unique English prompts in the Search Arena dataset. We generate prompt embeddings using OpenAI's `text-embedding-3-large` model, perform dimensionality reduction and clustering via UMAP and HDBSCAN, and summarize each resulting cluster using GPT-4o (Tang et al., 2025). We adapt the Arena Explorer (Tang et al., 2025) methodology for the Search Arena dataset. Figure B.4 illustrates the distribution of topics derived from BERTopic clustering

over the Search Arena prompts. The prominence of categories such as Technology Comparisons, Market Analysis, and Entertainment Characters, alongside a long-tail of niche domains, highlights the dataset's breadth and its suitability for evaluating search-augmented LLMs under diverse topics.

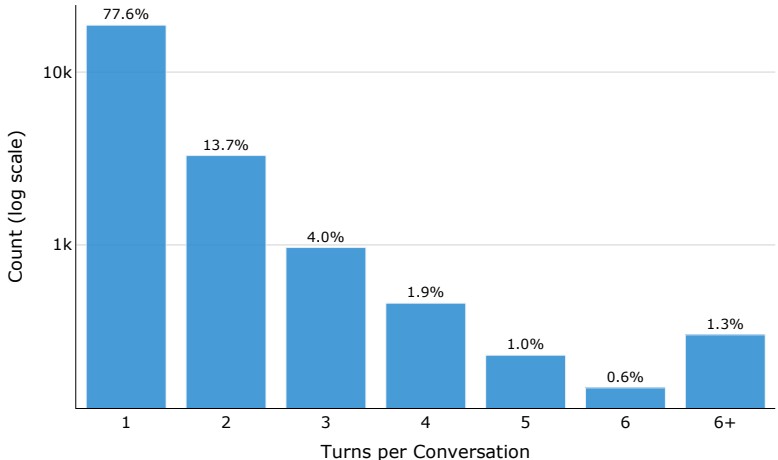

Figure B.3: **Search Arena Conversation Length (Number of Turns) Distribution.** Search Arena chats are multi-turn with 22.4% of conversations containing more than 1 turn.

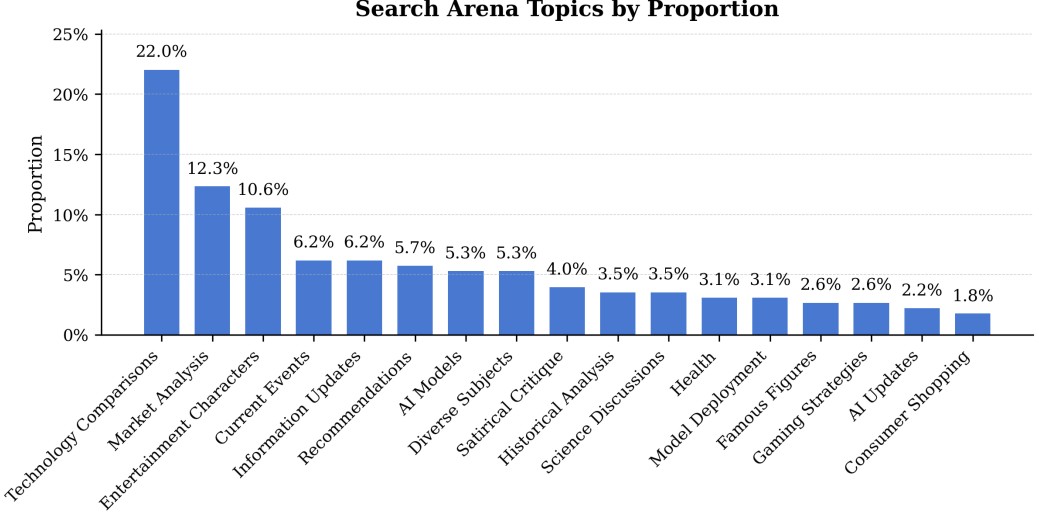

Figure B.4: **Top Topic Categories in Search Arena.** This figure shows the distribution of topic clusters. The most prevalent topics include *Technology Comparisons* (22.0%), *Market Analysis* (12.3%), and *Entertainment Characters* (10.6%). Less frequent but still diverse topics include health and shopping. The long-tail distribution reflects the breadth of real-world usage of search-augmented LLMs.

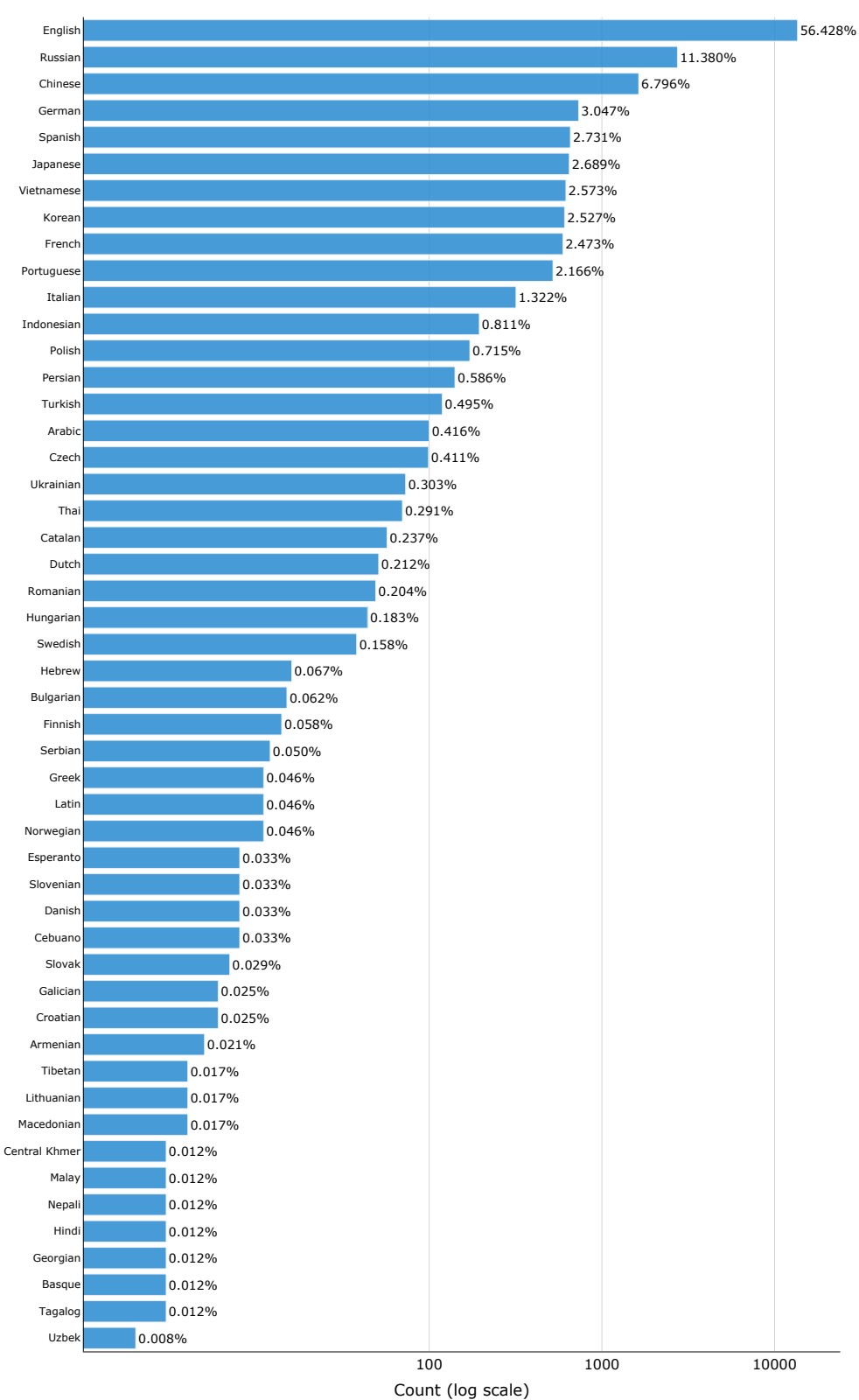

Figure B.5: **Search Arena Language Distribution (top 50).** Search Arena prompts span 71 languages.

## C DESCRIBING DIFFERENCES

To extract interpretable differences across two text corpora (e.g., prompts, responses), we use LLM-based dataset differencing methods (Zhong et al., 2022; Dunlap et al., 2024).

### C.1 PROMPT DIFFERENCES

To describe differences in prompt distributions between Search Arena and other datasets (see analysis in Subsection 2.2), we use GPT-4.1 to propose properties (hypotheses) across 16 rounds, with 32 samples per group in each round. We then use o3 to filter the top five properties and GPT-4.1-mini to re-rank them. The prompts are shown below. We use the default temperature parameter of 1.0.

We extract the following properties (p < 0.001) for each dataset pair and report the proportion of validation set examples (100 samples) satisfying each property.

**Search Arena vs SimpleQA:**

- Requests in-depth explanations, analyses, or step-by-step guidance rather than single factual answers (28% vs 0%).
- Requests recent or real-time information, including current events, product features, or online research (35% vs 7%).
- Seeks technical help, troubleshooting, or comparative evaluations for software, programming, or digital tools (17% vs 1%).
- Asks for creative content generation, rewriting, or stylistic transformations (stories, poems, satirical pieces, etc.) (13% vs 0%).

**SimpleQA vs Search Arena:**

- Requests exact factual details such as specific names, dates, numbers, or titles (97% vs 41%).
- Expects a single objective, verifiable answer rather than explanations or analysis (97% vs 43%).
- Avoids subjective, open-ended, or creative requests, limiting queries to factual retrieval (99% vs 52%).
- Focuses on niche or lesser-known historical, scientific, or cultural topics (51% vs 7%).
- Presents concise, narrowly scoped questions with minimal background information (91% vs 56%).

**Search Arena vs BrowseComp:**

- User messages are brief and provide minimal contextual detail (80% vs 23%).
- Responses expected are immediate functional outputs such as lists, summaries, or code snippets (50% vs 20%).
- Prompts seek practical advice or step-by-step instructions for real-world tasks (29% vs 1%).
- Queries focus on well-known, mainstream topics, products, or services (75% vs 48%).

**BrowseComp vs Search Arena:**

- Frames the query as a deductive puzzle or investigative challenge (80% vs 3%).
- Provides explicit temporal or geographic constraints that narrow the solution space (99% vs 25%).
- Requires synthesizing multiple detailed clues from different sources to deduce the answer (92% vs 36%).

**Search Arena vs Text Arena:**

- Requests for current factual information about real-world entities, events, or products (61% vs 25%).

- Requests for technical comparisons, evaluations, or purchasing guidance on devices, software, or services (16% vs 4%).

**Text Arena vs Search Arena:**

- Seeks analytical, step-by-step solutions to mathematical, logical, or technical puzzles and explanations (21% vs 4%).
- Requests programming assistance, such as debugging code, generating scripts, or explaining programming concepts (23% vs 8%).
- Requests creative writing outputs, including stories, poems, jokes, or fictional scenarios (20% vs 5%).

---

💬 **LLM Prompt for Proposing Distinguishing Properties between Prompt Sets**

```
The following are a two separate lists of prompts that users have
asked to a chatbot:
<START>
{text}
<END>

I am a machine learning researcher trying to figure out the major
differences between these two groups of prompts.  This is a very small
portion of the data, so I want the differences to be general.

Please provide a list of the top three differences (separated by
bullet points "*") between the prompts in Group A and Group B.

Follow the detailed instructions below:
- Identify properties that are more common in Group A prompts compared
to Group B prompts.
- The property descriptions should be simple and detailed.  Do not
produce vague and generic descriptions.
- Start the description of each property with "User".  Do not use
keywords like "group A", "group B", "more common", "less common",
"frequently", "occasionally" in your response.
- Do not combine multiple properties / features in a single bullet
point.
- An example of the desired output format:
* "Property 1"
* "Property 2"
* "Property 3"

Please order your response in terms of the most common differences
between the two groups.  Your response:
```

---

💬 **LLM Prompt for Reducing Proposed Properties**

```
PROPERTIES:
##############################
{properties}
##############################

Above is a list of properties, several of which are similar.
- Please reduce this to a list of the five most common distinct
properties, ordered by frequency of occurrence.
- There should be no similar / overlapping properties in the final
list.
- Do not group multiple unrelated properties / features into a single
property.
- Do not use keywords, like "frequent", "rare", "common", "uncommon",
etc.
- Only respond with a numbered list of properties, no other text.
```

> **💬 LLM Prompt for Ranking Properties**
>
> ```
> PROMPT:
> <START>
> {text}
> <END>
>
> PROPERTY:
> <START>
> {hypothesis}
> <END>
>
> Check whether the PROMPT satisfies the PROPERTY. Respond with Yes or
> No.  If you are unsure, respond with No.
> Output:
> ```

## C.2  RESPONSE DIFFERENCES

To describe differences in responses between search and non-search models in cross-arena deployments (see analysis in Subsection 3.3), we use GPT-4.1 to propose properties (hypotheses) across 32 rounds, with 8 samples per group in each round. We then use o3 to filter the top five properties and GPT-4.1-mini to re-rank them. The prompts are shown below (for property reduction, we re-use the same prompt from Subsection C.1). We use the default temperature parameter of 1.0.

For each dataset pair, we extract the top properties and report the proportion of validation set examples (100 samples) satisfying the properties, along with corresponding significance levels (p-values).

**Text Arena Setting**

Search vs non-search model responses to *Factual Lookup* prompts:

- Includes precise quantitative or technical specifics such as exact dates, numerical values, specifications, or code snippets (53.8% vs 41%) ($p = 0.26$).

Search vs non-search model responses to *Info Synthesis* prompts:

- Incorporates precise data, statistics, dates, names, and domain-specific terminology (80.1% vs 61.9%) ($p = 0.18$).

Non-search vs search model responses to *Text Processing* prompts:

- Uses explicitly structured formatting (numbered or bulleted lists, headings) to organize information (74.4% vs 56.4%) ($p = 0.1$).

**Search Arena Setting**

Search vs non-search model responses to *Factual Lookup* prompts:

- Provides extensive background context or explanatory preamble before addressing the specific question (83.3% vs 72.2%) ($p = 0.26$).

Search vs non-search model responses to *Info Synthesis* prompts:

- Explicitly states limitations, uncertainties, or caveats about the information provided (71.7% vs 52.2%) ($p = 0.05$).

---

💬 **LLM Prompt for Proposing Distinguishing Properties between Model A and Model B Responses**

```
The following is a list of Model A and Model B outputs to a set of
user questions:
<START>
{text}
<END>

I am a machine learning researcher trying to figure out the major
differences between the outputs of model A and model B. This is a very
small portion of the data, so I want the differences to be general.

Please provide a list of three top differences (separated by bullet
points "*") between the outputs of model A and model B.

Follow the detailed instructions below:
- Identify properties that are more common in model A outputs compared
to model B outputs.
- The property descriptions should be simple and detailed.  Do not
produce vague and generic descriptions.
- Do not use keywords like "model A", "model B", "more common", "less
common", "frequently", "occasionally" in your response.
- Do not combine multiple properties / features in a single bullet
point.
- An example of the desired output format:
* "Property 1"
* "Property 2"
* "Property 3"

Please order your response in terms of the most common differences
between the two groups.  Your response:
```

---

💬 **LLM Prompt for Ranking Properties**

```
QUESTION:
<START>
{question}
<END>

ANSWER:
<START>
{answer}
<END>

PROPERTY:
<START>
{hypothesis}
<END>

Check whether the ANSWER satisfies the PROPERTY. Respond with Yes or
No.  If you are unsure, respond with No.
Output:
```

## D   Leaderboard and General Feature Analysis

### D.1   Leaderboard Analysis

Models supported on the Search Arena platform are shown in Table A.1. Number of battles across models is shown in Figure D.1; pairwise battle count is shown in Figure D.2.

Average win rates across models are shown in Figure D.3. Pairwise win rates are shown in Figure D.4.

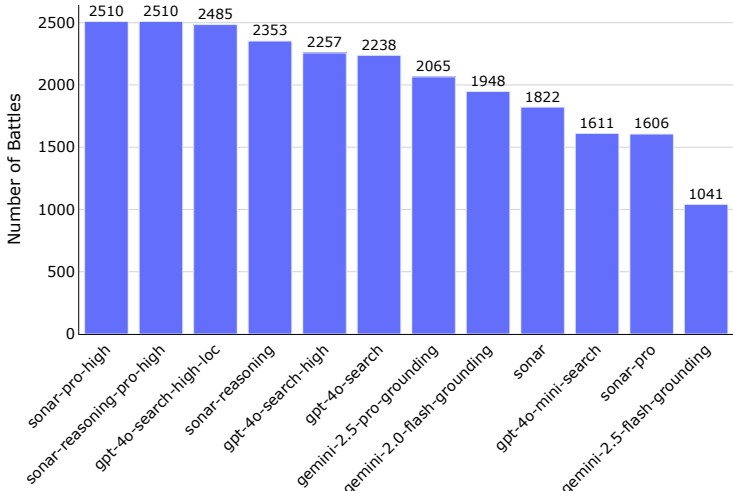

Figure D.1: **Battle Count Distribution**. Number of battles across Search Arena models. The distribution is not even due to (1) different sampling weights and (2) models were not added to the platforms at the same time.

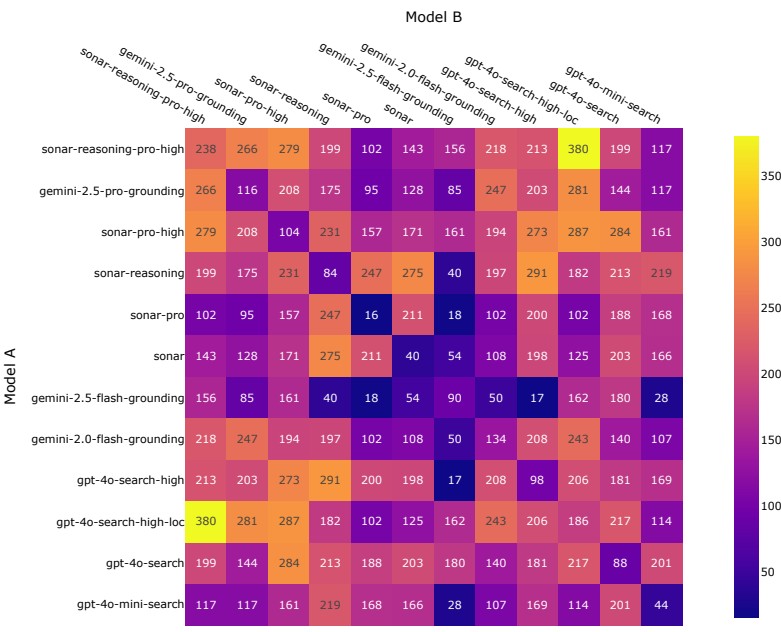

Figure D.2: **Pairwise Battle Count Distribution**. Number of battles between Search Arena models.

Consistent with Chatbot Arena's ranking system (Chiang et al., 2024), we use Bradley-Terry regression (Bradley & Terry, 1952) to calculate model coefficients and then re-scale to match the scale of the Elo rating system. The resulting model scores are shown in Figure D.5.

Search Arena model ratings and ranks are shown in Table D.1. We also compute model scores and the leaderboard for two subsets of the full dataset:

- English and non-English prompts: The gap between the two top models increases, with `sonar-reasoning-pro-high` performing better on non-English prompts. Additionally, `gemini-2.0-flash-grounding` and `gemini-2.5-flash-grounding` perform better on non-English prompts.

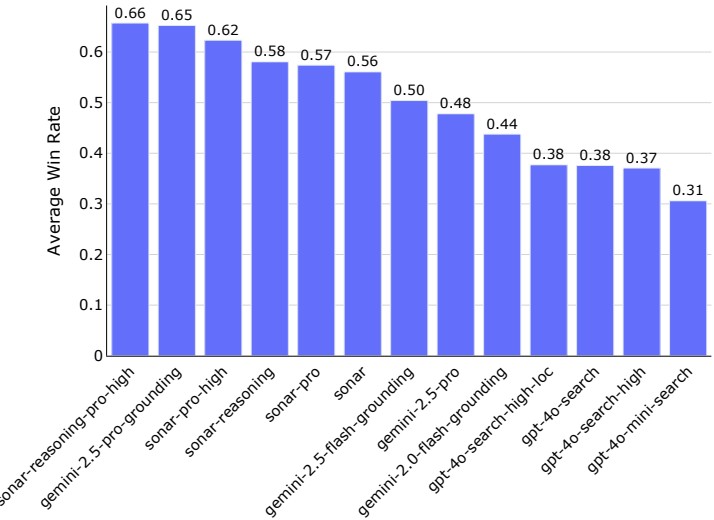

Figure D.3: **Win Rate Distribution**. Average win rates of Search Arena models.

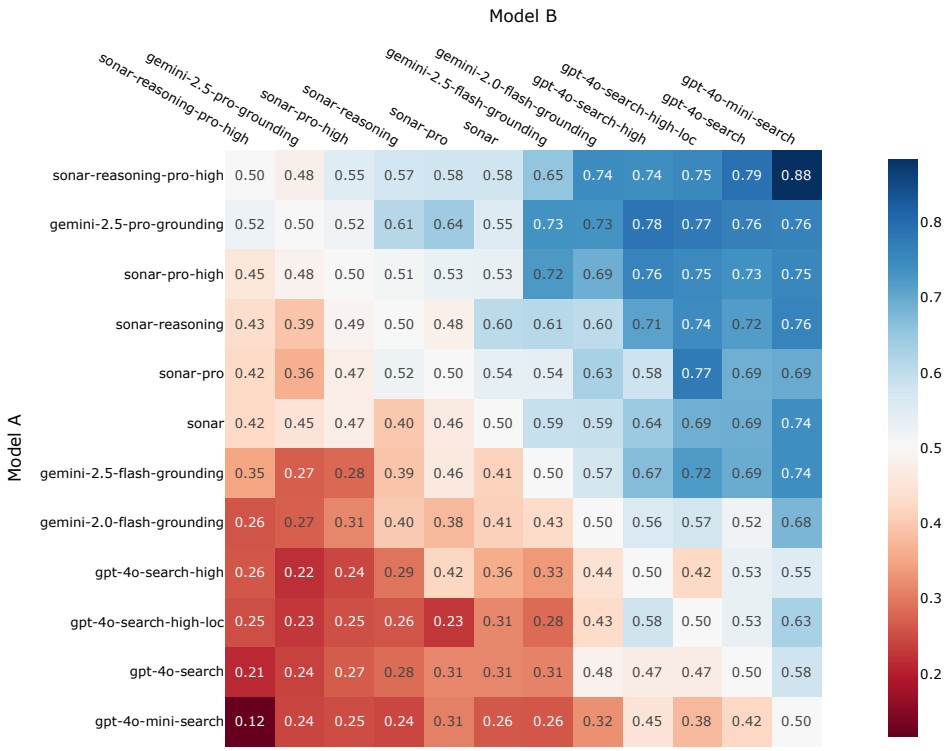

Figure D.4: **Pairwise Win Rates**. Pairwise win rates between Search Arena models.

- Factual and non-Factual prompts: We observe a clear split in the leaderboard on the Factual subset. Additionally, the gap between the three top models (including `sonar-pro-high`) "zeros out" on the non-factual subset.

## D.2 RESPONSE LENGTH ANALYSIS

Average response length distribution across Search Arena models is shown in Figure D.6 (Left). Reasoning models tend to be more verbose except for `sonar-reasoning`. `sonar-pro`'s version

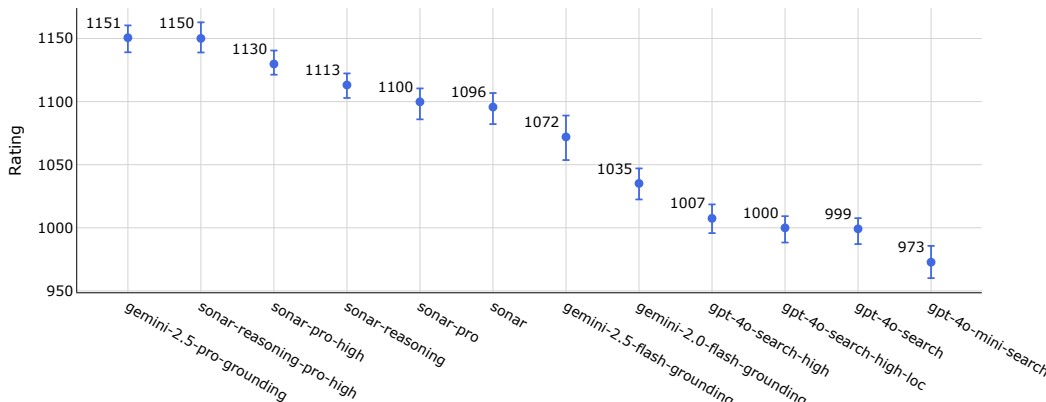

Figure D.5: **Search Arena Leaderboard**. Model scores based on Elo-scaled Bradley-Terry coefficients.

Table D.1: **Search Arena Leaderboard.** Elo-scaled Bradley-Terry ratings along with corresponding ranks (Rating (Rank)). Model ratings based on two subsets of the full dataset: (1) English vs non-English prompts, (2) Factual (*Factual Lookup* and *Info Synthesis*) vs non-Factual prompts.

| Model | Rating | Rating (English) | Rating (non-English) | Rating (Factual) | Rating (non-Factual) |
|---|---|---|---|---|---|
| gemini-2.5-pro-grounding | **1150.6 (1)** | **1146.8 (1)** | 1158.3 (1) | 1155.8 (1) | 1143.9 (1) |
| sonar-reasoning-pro-high | 1150.1 (1) | 1137.5 (1) | **1174.2 (1)** | **1161.4 (1)** | 1142.5 (1) |
| sonar-pro-high | 1129.8 (1) | 1125.7 (1) | 1136.2 (2) | 1110.1 (3) | **1144.1 (1)** |
| sonar-reasoning | 1113.2 (3) | 1112.0 (2) | 1117.9 (3) | 1105.9 (3) | 1121.0 (2) |
| sonar-pro | 1099.9 (4) | 1101.5 (3) | 1093.1 (4) | 1092.0 (3) | 1106.8 (4) |
| sonar | 1095.7 (4) | 1092.8 (4) | 1100.6 (3) | 1085.6 (3) | 1106.0 (4) |
| gemini-2.5-flash-grounding | 1072.0 (5) | 1061.4 (6) | 1098.4 (3) | 1095.0 (3) | 1059.2 (7) |
| gemini-2.0-flash-grounding | 1035.1 (8) | 1017.8 (8) | 1060.7 (5) | 1035.5 (8) | 1035.4 (7) |
| gpt-4o-search-high | 1007.5 (9) | 1000.9 (8) | 1018.7 (8) | 1007.7 (8) | 1005.6 (8) |
| gpt-4o-search-high-loc | 999.9 (9) | 989.9 (8) | 1019.7 (9) | 987.1 (9) | 1009.4 (8) |
| gpt-4o-search | 999.2 (9) | 999.4 (8) | 1002.1 (9) | 1000.5 (8) | 999.5 (9) |
| gpt-4o-mini-search | 972.7 (12) | 970.7 (9) | 973.6 (11) | 973.1 (9) | 969.6 (11) |

with a higher search context generates longer responses compared to the version with a medium context. Response length distribution is uniform across OpenAI's models.

We control for response length difference in the Bradley-Terry model and compute the corresponding coefficient across different subsets of the full data, broken down by intent category (see Figure D.7). For all intent categories, the coefficient (effect) is statistically significant; however, as expected, the effect is smallest for *Factual Lookup* prompts.

# E CITATION ANALYSIS

## E.1 CITATION COUNT

Citation count distribution is shown in Figure D.6 (Right). As expected, models with higher search context size cite more sources (e.g., `sonar-pro-high` vs `sonar-pro`). Furthermore, reasoning models tend to cite less sources compared to non-reasoning variants (e.g., `sonar-reasoning-pro-high` vs `sonar-pro-high`, `sonar-reasoning` vs `sonar`). We hypothesize that reasoning models synthesize and filter irrelevant sources before final response generation, resulting in less cited sources in the final response (see Figure 3). Interestingly, `gemini-2.5-pro-grounding` cites fewer sources compared to `gemini-2.5-flash-grounding`, suggesting that even though both are reasoning models, `gemini-2.5-pro-grounding` filters out more sources from the final response.

We then control for citation count difference in the Bradley-Terry model and compute the corresponding coefficient across different subsets of the dataset broken down by intent category (see Figure E.1). The effect of citation count on *Guidance* (e.g., debugging, problem-solving) prompts is not significant. The effect is largest on *Analysis* prompts.

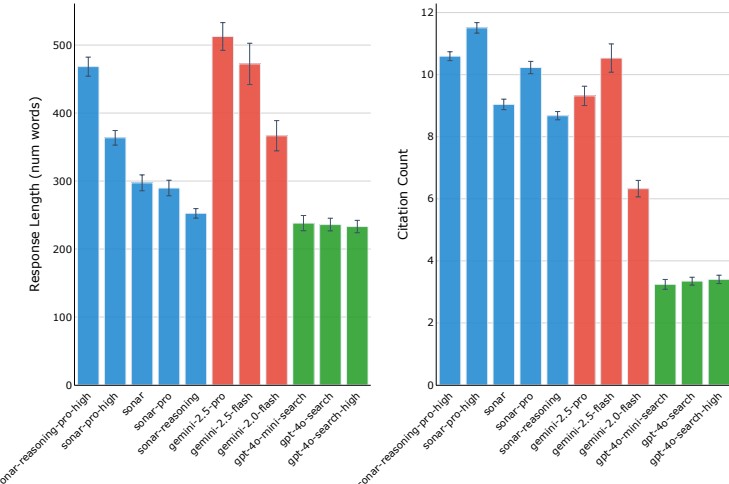

Figure D.6: **(Left)** Response length distribution. Reasoning models and models with high search context size tend to be more verbose. **(Right)** Citation count distribution. As expected, models with higher search context size cite more sources. Reasoning models cite fewer sources compared to non-reasoning variants (e.g., `sonar-reasoning-pro-high` vs `sonar-pro-high`).

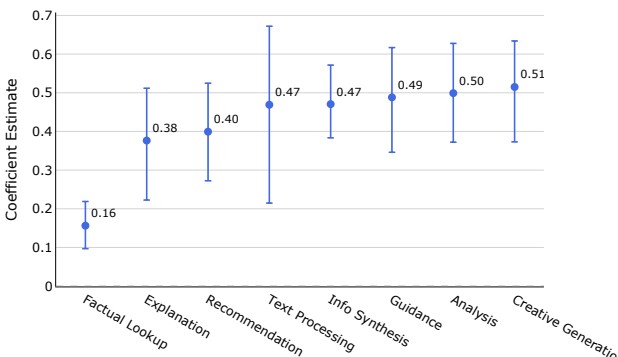

Figure D.7: **Response Length Control across Intents**. Bradley-Terry coefficients corresponding to response length across different intent categories. Length has less effect on *Factual Lookup* prompts.

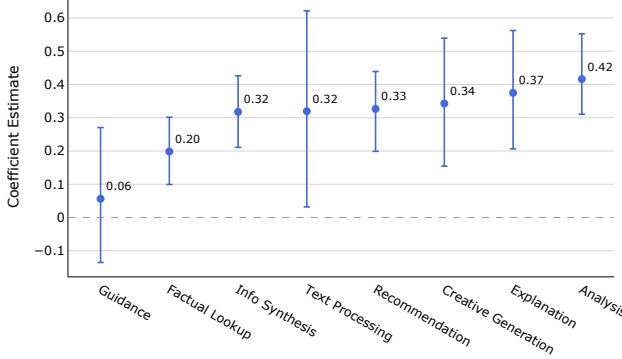

Figure E.1: **Citation Count Control across Intents**. Bradley-Terry coefficients corresponding to citation count across different intent categories. Citation count does not have significant effect on *Guidance* (e.g., debugging, problem-solving) prompts. The effect is largest for *Analysis* queries.

## E.2 CITATION SOURCES

To categorize citation sources, we use the following mapping from source domains to categories:

- **youtube**: "youtube.com".
- **gov_edu**: ".gov", ".edu", ".mil".
- **wiki**: "wikipedia", "wikihow", "wikimedia".
- **us_news**: "cnn.com", "apnews.com", "cnbc.com", "bloomberg.com", "economist.com", "nytimes.com", "washingtonpost.com", "wsj.com", "nbcnews.com", "abcnews.go.com", "usatoday.com", "npr.org", "latimes.com", "vox.com", "huffpost.com", "ft.com", "foxnews.com", "axios.com", "time.com", "buzzfeed.com", "cbsnews.com", "politico.co", "newsweek.com", "fortune.com", "theatlantic.com", "whattowatch.com", "scripp-snews.com", "investopedia.com", "yahoo.com", "breitbart.com", "washingtontimes.com", "dailycaller.com", "thefederalist.com", "townhall.com", "pjmedia.com", "westernjour-nal.com", "forbes.com".
- **foreign_news**: "reuters.com", "bbc.com", "aljazeera.com", "dw.com", "france24.com", "as.com", "elpais.com", "cbc.ca", "theglobeandmail.com", "smh.com.au", "abc.net.au", "japantimes.co.jp", "straitstimes.com", "hindustantimes.com", "thehindu.com", "economic-times.indiatimes.com", "indianexpress.com", "independent.co.uk", "theguardian.com", "ca-denaser.com", "lemonde.fr", "vnexpress.net", "ndtv.com".
- **social_media**: "tiktok.com", "facebook.com", "instagram.com", "x.com", "twitter.com", "linkedin.com", "snapchat.com", "pinterest.com".
- **community_blog**: "reddit.com", "quora.com", "blog", "medium.com", "wordpress.com", "substack.com", "tumblr.com".
- **tech_coding**: "github.com", "gitlab.com", "stackexchange.com", "microsoft.com", "dev.to", "codecademy.com", "stackoverflow.com".
- **academic_journal**: "jstor.org", "springer.com", "sciencedirect.com", "nature.com", "arxiv.org", "researchgate.com", "biorxiv.org".
- **retail**: "amazon.com", "ebay.com", "walmart.com", "target.com", "bestbuy.com", "costco.com".
- **other**: non-matched domains.

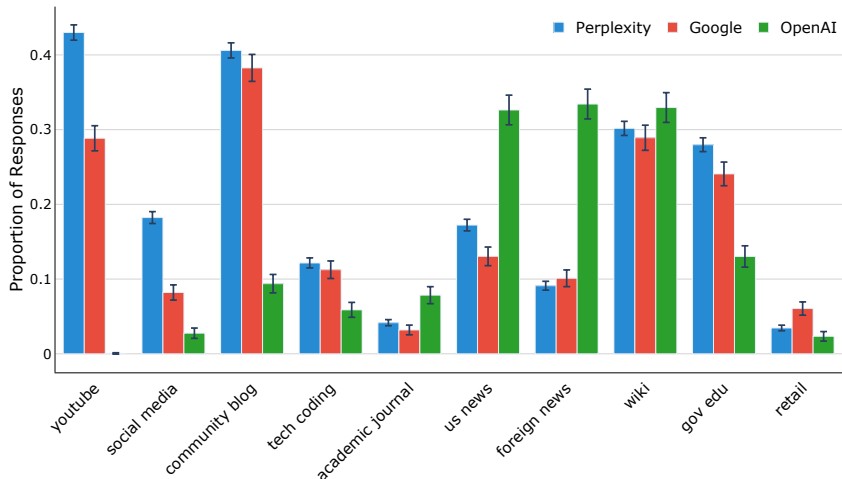

Figure E.2: **Citation Source Distribution across Model Families.** Models from different providers are biased towards different types of sources: (1) Perplexity models prefer citing YouTube, social media, and community blogs, (2) OpenAI models are biased towards mainstream news outlets.

The distribution of citation domain categories is shown in Figure E.2. Perplexity's models tend to cite social media (including YouTube) and community platforms (e.g., Reddit, Quora). OpenAI's GPT-4o-based models are more biased towards mainstream news outlets. Google's Gemini models are

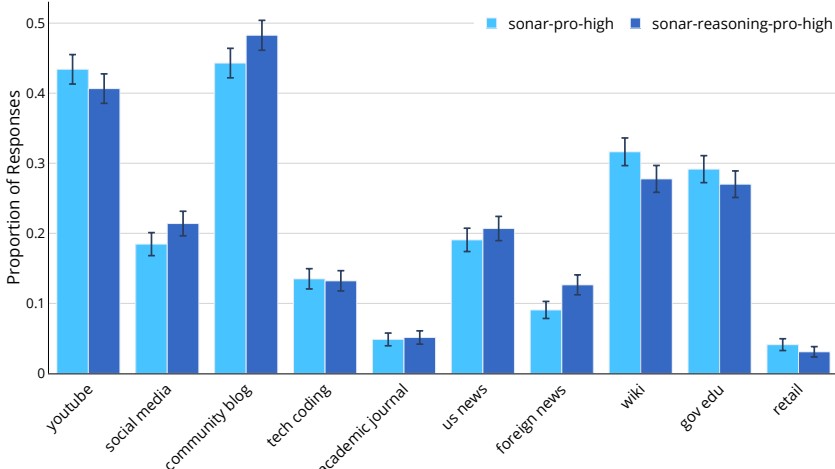

Figure E.3: **Citation Source Distribution across Non-reasoning (sonar-pro-high) and Reasoning (sonar-reasoning-pro-high) Models.** Although both systems are based on the same search module, the reasoning model (dark blue) cites more community and blog sources and less Wikipedia articles compared to the non-reasoning variant (light blue).

in between. To further understand the effect of reasoning, we analyze the distributional differences of the cited sources by `sonar-pro-high` (non-reasoning) and `sonar-reasoning-pro-high` (reasoning), as reported in Figure E.3. We find that even though both systems are based on the same search module, the reasoning model cites more community and blog sources and less Wikipedia articles, compared to the non-reasoning model. These results are consistent with our correlational analysis of user preferences and cited domains (e.g., users rejecting Wikipedia sources in specific settings).

We can also control for all features simultaneously (response length, number of citations, and citation sources) to compute adjusted model scores. The scores and rankings before and after applying these controls are shown in Figure E.4. We observe that the model scores and rankings tend to converge after the controls are applied. This convergence is particularly evident within model families: the top three models from Perplexity show significant convergence. This suggests that much of the variation between models in the same family is accounted for by the control features. We note that features such as factuality and coverage may be correlated with the number of citations or specific citation sources; we leave this analysis to future work.

### E.3 CITATION ATTRIBUTION ANALYSIS

As described in Subsection 3.2 and illustrated in Figure 7, we analyzed citation attribution on approximately 100 conversations per intent category. This section outlines the full pipeline.

We first sampled around 100 conversations from each intent category, excluding the "Other" class. To retrieve the cited content, we used Firecrawl[1] as our default batch scraping tool. For social media platforms or other domains where Firecrawl could not be applied, such as Reddit and YouTube, we leveraged official APIs or extracted video transcripts when available. For the remaining URLs that failed due to technical limitations, access restrictions, or licensing concerns, we excluded the corresponding samples. Overall, we took care to ensure ethical data usage, relying only on publicly accessible content and following best practices regarding rate limits and terms of service. In total, we collected over 20,000 web documents across 780 conversations.

Next, we used `gemini-2.5-flash-preview-0417` to analyze the data, selected for its strong performance on long-context reasoning, reliable structured (JSON) outputs, and reasonable cost. For each conversation, we first used the model to parse user messages into `(claim, URL)` pairs, following the structure shown in Figure 7(a). Then, for each pair, we provided the scraped markdown content and the associated claim to infer attribution judgments. Each attribution was labeled as `Support`, `Irrelevant`, or `Contradict`. The prompt used for LLM-based tagging is shown on the next page.

---

[1]https://www.firecrawl.dev

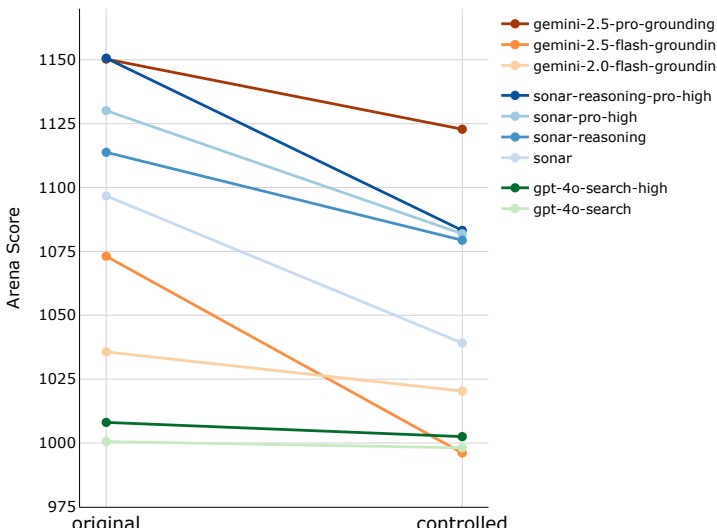

Figure E.4: **Model Scores Before and After Control.** Model scores and rankings converge after the controls are applied.

As illustrated in Figure 7(b), we aggregated the number of supporting, irrelevant, and contradicting claims per turn and used these counts as features in the Bradley-Terry model, following the same setup as the other control experiments. Human experts validated a subset of the outputs. Future work can further scale this pipeline or add onto analysis methodology.

---

💬 **LLM Prompt for Claim–Citation Attribution Assessment**

```
Task:  Tell me if the claim can be verified or supported by the web
content.
Specifically:

• If the claim is supported by the web content, return "support"

• If the claim is contradict to the web content, return "contradict"

• If the claim is completely irrelevant to the web content, return
  "irrelevant"

Claim:  {claim}
Web content:  {web content}

Return in the json format:

{
    "reasoning": "<explain the reasoning>",
    "answer": <"support" | "contradict" | "irrelevant">
}
```

---

## F    CROSS-SETTING ANALYSIS

In this section, we extend our analysis of Subsection 3.3 and study model performance changes across different benchmarks. Specifically, we explore the following research question: *How does model performance and ranking change across different search and non-search benchmarks?*

For offline evaluation of models in search settings, we use BrowseComp (Wei et al., 2025) and SimpleQA (Wei et al., 2024)[2]. For testing the general performance of the models in non-search settings, we use ArenaHard-v2 (Li et al., 2024). BrowseComp and SimpleQA assess factuality based

---

[2]Due to the high cost of running search-augmented LLMs, we evaluated each model on the same randomly sampled subset of 500 questions. The subset was sampled once and shared across all models.

on well-specified ground truths, while ArenaHard-v2 utilizes an LLM-as-a-judge framework on a set of challenging prompts filtered from Chatbot Arena's Text Arena dataset. Although these benchmarks test different aspects of standard and search-augmented LLMs, all are partially captured by the Search Arena prompt distribution shown in Figure 1.

The selected models have near-zero accuracy on the BrowseComp benchmark; this finding is not surprising, as the questions are specifically designed to evaluate and challenge deep research pipelines. The model scores for the rest of the benchmarks–Search Arena, SimpleQA, and ArenaHard-v2–are provided in Table F.1. Additionally, we calculate model win rates on two subsets of the Search Arena dataset based on the annotated intent classes (see Section 2)–Search Arena (Fact+Synth) includes only *Factual Lookup* and *Info Synthesis* queries, while Search Arena (Other) contains the remaining subset. We use Kendall's tau for comparing ranks and Pearson correlation for comparing raw scores across benchmarks. On SimpleQA, model accuracy is saturated (ranging from 89% to 93%), with minimal separability between the models and low agreement with Search Arena ($\tau = 0.422$, $r = 0.582$). In contrast, ArenaHard-v2 shows greater performance variance and separability, and has a higher agreement ($\tau = 0.556$, $r = 0.844$) with Search Arena compared to SimpleQA. However, the rankings differ; notably, the three reasoning models rank lower in ArenaHard-v2, suggesting that, while combining web search with reasoning improves performance in Search Arena, it may degrade performance on ArenaHard-v2 prompts. Additionally, when comparing the two subsets of the Search Arena with ArenaHard, Search Arena (Other) has higher agreement ($\tau = 0.644$, $r = 0.882$) compared to Search Arena (Fact+Synth) ($\tau = 0.511$, $r = 0.675$). This finding is expected as the prompt distribution in Search Arena (Other) is closer to that of ArenaHard (e.g., creative writing, problem-solving).

Furthermore, we compare model performance with and without search on the SimpleQA and ArenaHard-v2 benchmarks. We used Gemini models in this case study, as search is implemented as a tool (`GoogleSearch` tool) and can be easily turned on and off. The performance change is shown in Figure F.2. Search significantly improves model performance on SimpleQA (models' performance saturates at around 90%). However, performance degrades on ArenaHard-v2, with non-search variants achieving higher scores.

Table F.1: Model scores across three benchmarks–Search Arena (win rate), SimpleQA (accuracy), and ArenaHard-v2 (win rate). On SimpleQA, models' performance is saturated and lead to minimal separability in model scores. Search Arena and ArenaHard-v2 provide more separability, but the rankings are different.

| Model | Search Arena | Search Arena (Fact.+Synth.) | Search Arena (Other) | SimpleQA | ArenaHard-v2 |
|---|---|---|---|---|---|
| sonar-reasoning-pro-high | **66.6 (-2.1 / +2.2)** | **68.8 (-3.3 / +3.4)** | 65.0 (-3.1 / +2.7) | 91.8 (-2.4 / +2.4) | 28.3 (-1.7 / +2.1) |
| gemini-2.5-pro-grounding | **66.7 (-2.5 / +2.4)** | 68.3 (-3.8 / +3.7) | 65.3 (-3.2 / +3.2) | 90.6 (-2.6 / +2.4) | 33.5 ( -2.1 / +2.4) |
| sonar-pro-high | 63.7 (-2.2 / +2.4) | 59.9 (-3.5 / +3.3) | **66.7 (-2.8 / +2.9)** | **93.2 (-2.2 / +2.2)** | **43.4 (-2.1 / +2.2)** |
| sonar-reasoning | 60.1 (-2.5 / +2.3) | 58.8 (-3.4 / +3.3) | 61.3 (-3.2 / +3.4) | 92.6 (-2.4 / +2.2) | 29.1 (-2.4 / +2.3) |
| sonar-pro | 57.5 (-3.1 / +2.8) | 56.8 (-4.3 / +4.2) | 58.4 (-3.9 / +4.0) | 92.6 (-2.4 / +2.2) | 39.9 (-2.2 / +2.2) |
| sonar | 55.8 (-2.7 / +2.8) | 54.8 (-4.1 / +3.8) | 56.8 (-3.9 / +3.8) | 91.8 (-2.4 / +2.4) | 36.2 ( -1.9 / +2.1) |
| gemini-2.5-flash-grounding | 49.8 (-3.7 / +3.5) | 54.5 (-5.9 / +5.8) | 46.1 (-4.8 / +4.7) | 89.8 (-2.6 / +2.6) | 22.6 (-1.4 / +1.8) |
| gemini-2.0-flash-grounding | 41.7 (-2.8 / +2.9) | 42.8 (-4.2 / +4.2) | 41.0 (-3.5 / +3.4) | 89.2 (-2.8 / +2.6) | 13.4 (-1.0 / +1.1) |
| gpt-4o-search-preview-high | 34.8 (-2.5 / +2.5) | 36.2 (-3.6 / +3.8) | 33.6 (-3.5 / +3.2) | 89.4 (-2.8 / +2.6) | 16.6 (-1.2 / +1.2) |
| gpt-4o-mini-search-preview | 29.3 (-2.7 / +2.6) | 30.3 (-3.7 / +4.2) | 28.6 (-3.6 / +3.6) | 91.2 (-2.6 / +2.4) | 8.3 (-0.6 / +0.7) |

# G  ADDITIONAL COMPARATIVE ANALYSES

## G.1  CORAL

As our setup is similar to prior work in conversational search, we conduct additional comparative analysis with CORAL (Cheng et al., 2025a), a benchmark designed to assess RAG systems in multi-turn conversational settings. Compared to CORAL, Search Arena conversations are collected from natural open-ended interactions between humans and search-augmented LLMs and contain user-provided preference labels.

Similar to the datasets studied in the main paper, CORAL focuses on English-only conversations, whereas the Search Arena prompts cover a diverse set of 71 languages. Search Arena (24k) also is larger in scale compared to CORAL (8k). Furthermore, in contrast to SimpleQA (Wei et al., 2024) and BrowseComp (Wei et al., 2025) and similar to Search Arena, CORAL includes multi-turn conversations. Compared to Search Arena (Figure B.3), CORAL conversations are longer on average (Figure G.1, Left). In terms of prompts, CORAL focuses on information-seeking queries only,

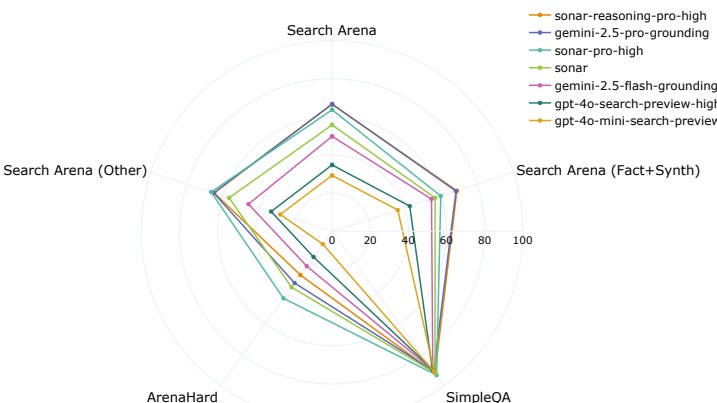

Figure F.1: **Model Scores Across Benchmark.** (1) Models' performance on SimpleQA is saturated with minimal separability. (2) Performance variance in ArenaHard-v2 is comparable to that of Search Arena, but the rankings are different, with reasoning models having higher performance in search settings. (3) Model scores and ordering in Search Arena (Other) is closer to that of ArenaHard compared to Search Arena (Fact+Synth).

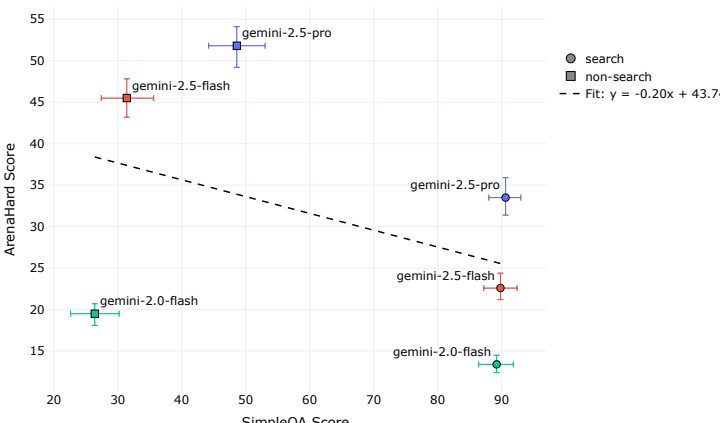

Figure F.2: **SimpleQA vs ArenaHard-v2 Performance**. Search and non-search performance of Gemini models on SimpleQA and ArenaHard-v2 benchmarks. Search improves performance on SimpleQA, while degrades the score on ArenaHard-v2.

whereas Search Arena covers a broader range of intents (e.g., factual lookup, creative generation). The prompt length distribution of Search Arena prompts is also more varied compared to CORAL, with the latter containing more concise questions (see Figure G.1, Right).

Additionally, in contrast to Search Arena, CORAL explicitly constructs ground truth answers, assuming a bounded set of documents (including golden). Search Arena focuses on human preferences, which, as observed in Section 3, do not need to correlate with factuality and quality of the retrieval. Finally, Search Arena data is collected from real-world natural interactions, enabling us to study real-world usage of the models and analyze how model response features correlate with the users' preferences. Thus, the differences between the two datasets are due to the differences in the corresponding intents, use cases, and motivations.

## G.2   WILDCHAT

Similar to the Text Arena and Search Arena datasets studied in the main paper, WildChat (Zhao et al., 2024) is a crowd-sourced human-LLM conversation dataset. Similar to the Text Arena and in contrast to the Search Arena, the interactions in WildChat are with regular LLMs without access to additional tools, such as web search.

Despite the differences in the settings, we compare WildChat with Search Arena across multiple axes; we use a random sample (100k) of the non-toxic filtered version (3.2M) of the original dataset for the comparative analysis below. First, the full WildChat dataset is much larger in scale (4.8M), compared to Search Arena (24k). Both datasets are multilingual, containing interactions with over 70 languages. Turn and prompt length distributions of the WildChat dataset are provided in Figure G.2. WildChat turn count distribution is similar to that of Search Arena, with Search Arena containing slightly more multi-turn conversations. Additionally, WildChat prompts are longer on average (594.9) compared to those of Search Arena (57.1). This difference arises from the difference in the user intents across the two datasets: as reported in the Text Arena comparative analysis, search-enabled settings significantly impact the intent of the queries (e.g., users more likely to ask real-time information-seeking questions).

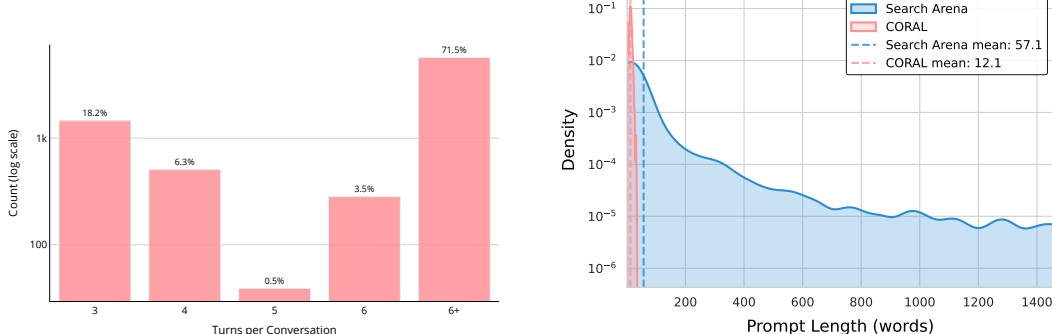

Figure G.1: **(Left)** CORAL conversation length (number of turns) distribution; CORAL conversations are longer than those in Search Arena, with 8.3 turns on average. **(Right)** CORAL prompt length distribution; CORAL prompts (red) are more concise and less diverse (lower variance) compared to Search Arena prompts (blue).

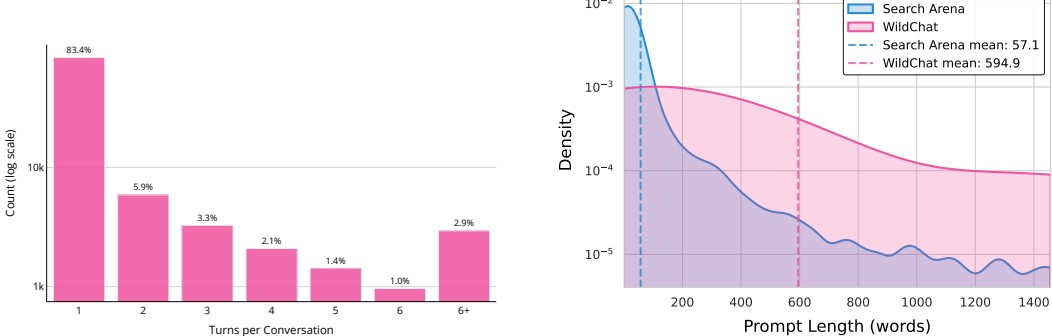

Figure G.2: **(Left)** WildChat conversation length (number of turns) distribution; WildChat conversations are similar in distribution to Search Arena Figure B.3, with slightly fewer multi-turn samples. **(Right)** WildChat prompt length distribution; WildChat prompts (pink) are longer compared to those of Search Arena (blue).

