# OpenReview forum: "Search Arena: Analyzing Search-Augmented LLMs"
_ICLR.cc/2026/Conference — ICLR 2026 Poster_

### Official Review · Reviewer_Tf64 · 2025-10-23

**Soundness:** 3
**Presentation:** 3
**Contribution:** 3
**Rating:** 6
**Confidence:** 4

**Summary:**

This paper introduces Search Arena, a large-scale, multilingual and multi-turn benchmark for search-augmented LLMs that contains 24k real user conversations and 12k human-preference judgements. The authors analyse how response length, number of citations, source types and citation fidelity correlate with user preference, and conduct cross-arena experiments to quantify the gain of web access. The dataset and ranking pipeline are released under CC-BY.

**Strengths:**

- The benchmark itself is a valuable contribution: it is two orders of magnitude larger than prior search-LLM datasets, covers 70+ languages, multi-turn context and nine diverse intents, and provides full system traces (query, retrieved pages, response, human vote).
- The quantitative analysis offers actionable insights—e.g. users reward more citations even when irrelevant, Wikipedia citations are penalised, and reasoning models filter sources more aggressively—all statistically validated with a Bradley-Terry model.

**Weaknesses:**

- As an evaluation benchmark the dataset is still tied to human preference. Automatic metrics are only sketched (e.g., claim-citation labels), so new models must either run costly human battles or risk LLM-as-a-judge drift. The paper also does not demonstrate any model-training experiments that exploit the released supervision.
- The main qualitative insights—citation count boosts perceived credibility, users distrust Wikipedia, etc.—largely echo existing insights; the work does not surface novel, counter-intuitive findings.
- Although ethics are discussed, the entire corpus is mined from users who clicked “accept” once. No right-to-withdraw, no IRB statement, and limited sensitive-content filtering leave lingering privacy concerns.

**Questions:**

- What concrete steps will you take to handle residual PII or sensitive queries that automatic filters missed, and will you offer users a mechanism to retroactively remove their data?
- For groups that wish to evaluate a new model on Search Arena without running fresh human battles, which fully automatic metric pipeline do you recommend, and how should statistical significance be computed when only LLM judges are available?

---

> ### Author Response · Authors · 2025-11-22
>
> Thanks for highlighting the contributions of our work, including the construction of a large-scale diverse realistic interaction dataset and the corresponding findings and insights. We would like to address the concerns below:
>
> ---
>
> **[W1, Q2] Clarification on Model Evaluation / Training**
>
> Thanks for the insightful comment and question. We would like to clarify the main goals and motivations behind our work. In this work, we aim to (1) collect a diverse set of human interaction traces with search-augmented LLMs, accompanied with human preference labels, (2) analyze these interactions, (3) study how different models features (e.g., citation count, citation sources) impact user preferences, and (4) extract insights for future model evaluation and development. Specifically, our goal is not evaluating and ranking models, but rather than understanding how humans interact with these models and how their preferences are shaped.
>
> For automatic large-scale labeling, we utilized LLM-as-a-judge in specific settings and pipelines; for example, we used an LLM-based annotation pipeline to label intent classes and classify citation-claim pairs for the citation attribution analysis. Regarding offline evaluation of models, we explicitly do not recommend treating existing LLM-as-a-judge setups as a drop-in proxy for human preference. Many queries in Search Arena are time-sensitive, and an offline judge may not have access to the same information state as an online search model, unless the relevant web content is carefully archived. We emphasize these challenges in the revised paper and view the design of robust, fully automatic metrics as a promising direction for future work (Lines 492-510).
>
> Regarding training experiments, training models falls outside the scope of our identified goals; we intentionally omit surface-level training experiments and instead focus on the main objectives of the paper described in the first paragraph. We believe that future work can take insights from our paper and dataset, and develop models by augmenting our collected preference signals with additional objective signals, such as factuality and citation relevance. We appreciate the reviewer’s suggestions and revised the Limitations and Future Work section accordingly (Lines 492-510).
>
> ---
>
> **[W2] (Unsurprising) Findings and Insights**
>
> We agree that some high-level observations (such as citation count influencing perceived credibility) align with prior intuitions. However, while “distrust of Wikipedia” may seem intuitive to some readers, other reviewers (ttYs and Uoa4) explicitly flagged it as counterintuitive, suggesting that this finding is not universally assumed. Additionally, we would like to clarify that most of our findings are quantitative and based on large-scale correlational analysis over the collected data (24k).
>
> ---
>
> **[W3, Q1] PII Data Filtering and Policies**
>
> We appreciate the reviewer’s concerns about privacy. Our data collection process includes several safeguards:
>
> - Users explicitly agree via an “accept” button. The full terms of service every user accepted before using the service is provided in Appendix A.1 of the original submission.
> - We apply automated PII/content filtering before finalizing the dataset. The released corpus only contains anonymized text without direct user identifiers (Lines 904-908).
>
> For residual PII or sensitive queries that automatic filters might miss, we have a documented mechanism for removal. Concretely, users can contact the platform provider to resolve PII issues and remove the requested data points from the dataset. We added a placeholder for this contact channel in Appendix A.1 (Lines 908-909) and intend to replace it with a non-anonymous link in the final version.

---

### Official Review · Reviewer_Uoa4 · 2025-10-31

**Soundness:** 2
**Presentation:** 3
**Contribution:** 3
**Rating:** 6
**Confidence:** 4

**Summary:**

The paper introduces Search Arena, a crowd-sourced dataset for analyzing search-augmented language models collected by deploying a chat platform to collect in-the-wild user interactions. The dataset contains >24k paired multi-turn conversations and over 12k human preference votes, collected from interactions with 13 different models. This work addresses limitations of existing factoid benchmarks that are focused on single-turn, fact-checking questions. The dataset spans 71 languages and presents a new user intent taxonomy.

The authors use this dataset to analyze user preferences and find that users prefer longer responses and more citations. An interesting finding is that this preference for more citations holds even when the cited content is irrelevant to the claim. The paper also presents a cross-arena analysis, where they show that search-augmented models perform comparably in general-purpose chat settings, but models relying only on parametric knowledge underperform in search-intensive settings.

**Strengths:**

- The paper is well-motivated and easy to follow. It addresses a clear gap, i.e. the lack of large-scale multi-turn and multilingual datasets for evaluating search-augmented LLMs.
- The Search Arena dataset can be very useful to researchers working on a broad set of topics. Its scale as well as diversity in languages and query intents make it a valuable resource.
- The analysis of user preferences is interesting and results in some non-obvious findings. The finding that user preference is positively correlated with the number of citations, even when those citations are irrelevant / do not support the claims, is concerning and could affect model design decisions.
- The cross-arena analysis is interesting as well because it provides strong evidence that search augmentation is beneficial in search-heavy tasks without degrading general chat performance.

**Weaknesses:**

- The paper presents a lot of correlational evidence, for eg, about which response features and cited sources users find important, but it states these findings more definitively than the correlational analysis supports. I think this is a major weakness and I would suggest rewriting these claims with caution.

- The paper relies heavily on LLM-based pipelines for analysis, particularly for user intent classification and citation attribution. For the citation attribution validation, the process is described vaguely. For eg, the paper states that "Human experts validated a subset of the outputs" but does not provide inter-annotator agreement scores or a quantitative comparison between the LLM pipeline and human judgments. Given that the finding about irrelevant citations is a significant takeaway, a more robust validation of this pipeline is necessary.

- Relatedly, the citation attribution pipeline was run on just 100 English conversations per intent category due to cost / scraping challenges, which is a pretty small subset. It is unclear if these findings about irrelevant citations generalize across all languages and the full dataset.

- The paper provides limited discussion of some relevant prior work, such as WildChat and WildBench.

- The paper offers limited to no discussion on precisely how future work should utilize the Search Arena dataset.

**Questions:**

- The authors mentioned that reasoning models filter irrelevant content and cite fewer sources. Did you find any difference in the types of sources cited by reasoning vs non-reasoning models?
- It is unintuitive to me that citing Wikipedia is negatively correlated with user preference. Since it is hard to control the prompt set while just varying the cited sources, I wonder if the coefficient differences between cited sources are actually meaningful to draw conclusions from?

---

> ### Author Response · Authors · 2025-11-22
>
> Thanks for acknowledging the contributions of our work, including the utility of the dataset, thorough analysis, and insights. We would like to address your concerns below:
>
> ---
>
> **[W1] Correlational Analysis**
>
> Thanks for the comment. Indeed, our current analysis is purely correlational and following your suggestion, we have explicitly clarified this at the beginning of the analyses section in the updated paper (Lines 213-214).
>
> ---
>
> **[W2, W3] Citation Attribution Experiment Details**
>
> Thanks for raising the concern about LLM pipeline validation. Regarding the citation attribution experiment, we provide additional details below.
>
> Analyses were conducted on roughly 100 examples per category, therefore for a total of 800 examples (Lines 287-289). We would like to clarify that the sample was not limited to English conversations only (we mentioned English only in the original submission). This leads to scraping 6,595 web sources (each response in a pair has 4.12 cited sources on average) and labeling 24,127 citation-claim pairs extracted from the generated responses. Afterwards, we utilized an LLM-based annotation pipeline to label claim-citation pairs. To validate the LLM labels, two human annotators labeled a sample of 60 claim-citation pairs balanced across the three classes. We report 83.64% and 77.27% human-human and human-LLM average agreement rates (support vs not support), respectively.
>
> ---
>
> **[W4] Relevant Work**
>
> Thanks for suggesting additional related work. We included WildChat (Line 442) in the original Related Work section and added WildBench to the revised version (Line 441). Furthermore, following the reviewer’s suggestion, we broadened the scope of our dataset study and included additional comparative analysis between Search Arena and WildChat in Appendix G.2 (Lines 1930-1943, Figure G.2).
>
> ---
>
> **[W5] Search Arena for Future Work**
>
> In Section 5 of the submission, we briefly discussed future research directions guided by our current work. We would like to elaborate on some of these point below. While our primary focus is dataset construction and interaction analysis, the Search Arena dataset facilitates the following directions:
>
> - Objective metrics: As detailed in Section 3, user preferences are often influenced by surface-level features, such as citation counts or source reputation, rather than the intrinsic quality of the cited sources and the generated content. Consequently, user-provided labels may diverge from objective standards like factuality, citation-claim consistency, and source specificity. Future work should expand the evaluation framework to include these objective metrics and analyze their correlation with human preference signals to better understand the gap between perceived credibility and actual groundedness.
> - Offline evaluation: Our current evaluation relies on an online human-in-the-loop setup, which, while robust, restricts the scalability of testing new models. Future work can leverage our dataset to construct automated benchmarks for offline evaluation. This could involve training reward models or LLMs that approximate human preferences, potentially combining them with the objective metrics discussed above to create a comprehensive scoring rubric.
> - Model development: Our cross-arena and preference analyses reveal that blindly augmenting models with search is not always optimal. Future research can utilize these insights to improve both the search module (e.g., better source filtering) and the backbone LLM (e.g., optimizing policies for when to trigger search versus relying on parametric knowledge).
>
> We added more thorough discussion of future work directions in the revised Limitations and Future Work section (Lines 492-510).

---

> > ### Author Response · Authors · 2025-11-22
> >
> > **[Q1] Reasoning Model Cited Domain Analysis**
> >
> > Thanks for the insightful question. To freeze the search system and study how reasoning impacts the final response and the corresponding cited sources, we study the cited domain distribution of Sonar-Pro-High and Sonar-Reasoning-Pro-High models. The results are summarized in the table below:
> >
> > | Model / Domain    | Sonar-Pro-High (CI) | Sonar-Reasoning-Pro-High (CI) |
> > |-------------------|---------------------|--------------------------------|
> > | YouTube           | 0.43 (0.021)        | 0.41 (0.021)                   |
> > | Social Media      | 0.18 (0.016)        | 0.21 (0.018)                   |
> > | Community blog    | 0.44 (0.021)        | 0.48 (0.021)                   |
> > | Tech / coding     | 0.14 (0.014)        | 0.13 (0.014)                   |
> > | Academic journal  | 0.049 (0.009)       | 0.051 (0.009)                  |
> > | US News           | 0.19 (0.017)        | 0.21 (0.017)                   |
> > | Foreign News      | 0.09 (0.012)        | 0.13 (0.014)                   |
> > | Wikipedia         | 0.32 (0.020)        | 0.28 (0.019)                   |
> > | Gov / Edu         | 0.29 (0.019)        | 0.27 (0.019)                   |
> > | Retail            | 0.04 (0.008)        | 0.03 (0.007)                   |
> >
> > We observe that while both models are based on the same search system, the distributions of cited domains are different: for example, the reasoning model cites more community / blog sources (e.g., Reddit) and less Wikipedia sources. These results are consistent with our correlational analysis of user preferences and cited domains (e.g., users rejecting Wikipedia sources in specific settings). We added this interesting finding to the revised Appendix E.2 (Lines 1726-1756, Figure E.3).
> >
> > ---
> >
> > **[Q2] Cited Domain Analysis / Wikipedia**
> >
> > The authors were also initially surprised by the finding that citing Wikipedia is negatively correlated with users’ preferences. However, after qualitative inspection of the corresponding examples, we characterized the cases where Wikipedia might not be the most appropriate source to cite. For example, as shown in Figure 3 (Right) in the original submission, the users prefer more real-time sources (e.g., news articles) for more time-sensitive queries. This finding makes sense as Wikipedia is updated less frequently and might not be accurate for more time-sensitive dynamic queries.
> >
> > Additionally, we would like to acknowledge that the analysis is observational and correlational; we faithfully report the point estimates with the corresponding confidence intervals. This reporting along with large-scale corpus (~12.5k labels) to some extent mitigates the mentioned concern. As already noted in our response to W1, we have carefully revised the text to avoid over-claiming and we thank the reviewer for prompting us to clarify this point.

---

### Official Review · Reviewer_ttYs · 2025-10-31

**Soundness:** 3
**Presentation:** 4
**Contribution:** 3
**Rating:** 6
**Confidence:** 3

**Summary:**

The study introduces the first large-scale, crowd-sourced dataset of human interactions with search-augmented LLMs. It analyzes how users engage with search-augmented LLMs and finds that user preferences are strongly influenced by citation quantity and source type (even when citations are irrelevant). A cross-arena experiment shows that web search improves model performance in both search and non-search settings, but models without search capabilities perform poorly when users expect real-time retrieval.

**Strengths:**

-The study is impressively comprehensive, drawing on a diverse dataset that spans 11,650 users across 136 countries, 13 models, and 70 languages.

-The authors conducted a series of insightful and relevant analyses on human preferences in search-augmented LLMs, such as the “Types of Cited Sources” section. The finding that Wikipedia citations correlate negatively with user preference (while social media and community sources correlate positively) is both counterintuitive and well-explained (i.e., Wikipedia’s breadth and outdatedness often reduce perceived relevance).

-Impactful work: This work releases a large-scale, human-preference dataset for search-augmented LLMs, which acts as a stepping stone for future research works to understand how users evaluate search-augmented LLMs.

-The paper is well-structure and well-motivated, with the intent diversity section drawing clear comparisons to prior datasets. The writing is polished, and the narrative is easy to follow.

**Weaknesses:**

The current premise of search arena relies on human preference signals, but the paper’s own findings cast doubt on whether these signals are reliable indicators of true search quality. In particular, the Citation Attribution analysis shows that users often fail to distinguish between supporting and irrelevant citations and tend to prefer responses with a higher number of citations (regardless of the validity)). Human preference may conflate perceived credibility with actual factual correctness, leading to rankings that reward citation quantity or presentation style rather than true retrieval or reasoning quality.

The authors could expand their Citation Attribution experiment (currently only limited to ~100 examples) as a more objective evaluation metric. In particular, metrics assessing (i) citation accuracy, (ii) citation relevance (semantic alignment between retrieved source and response content), (iii) recency (temporal freshness of cited information), and (iv) retrieval rarity or specificity (the model’s ability to surface rare but authoritative sources) could provide a more faithful measure for search-augmented LLM performance.

**Questions:**

The reasoning traces shown in Figure 3 are quite interesting (demonstrating multi-document analysis, filtering, and synthesis). Did the authors observe cases where reasoning models misinterpreted their sources? If so, how might such errors in reasoning be detected at a larger scale?

---

> ### Author Response · Authors · 2025-11-22
>
> Thanks for acknowledging the contributions and the impact of our work. We would like to address your concerns below:
>
> ---
>
> **[W1] Preference Reliability**
>
> Thanks for the insightful comment. We would first like to restate that the central goals of the paper are (1) collecting a diverse dataset of human interactions with search-augmented LLMs and (2) conducting a thorough analysis on how users interact with these models and how different features of the models’ responses correlate with users’ preferences. We would like to clarify that we do not treat the human-preferred response as the objectively “correct” one, but rather as the response better aligned with the user’s personal preferences. Therefore, the preferences are not necessarily indicative of the search quality; in fact, our citation attribution analysis is precisely intended to highlight this gap. Thus, the goal of the paper is not ranking models or evaluating the factuality / quality of the models. Our goal is to conduct an in-depth analysis of user preferences under search settings, identify gaps, and uncover insights useful for future system development and evaluation. We clarify the above in the revised Limitations and Future Work section (Lines 492-510).
>
> ---
>
> **[W2] Expanding Evaluation**
>
> Regarding the Citation Attribution analysis, we would like to clarify that the analyses were conducted on roughly 100 examples ***per category*** (Factual Lookup, Info Synthesis, Analysis, Recommendation, Explanation, Creative Generation, Guidance, Text Processing), therefore for a total of 800 examples (Lines 287-289). This leads to scraping 6,595 web sources (each response in a pair has 4.12 cited sources on average) and labeling 24,127 citation-claim pairs extracted from the generated responses.
>
> Furthermore, as mentioned in our response to W1, we focus and conduct in-depth analysis of human-AI interactions under search settings, as well as studying the effects of model features on user preferences. Expanding the set of features to include citation accuracy (factuality), relevance, recency, and rarity / specificity is a promising future direction. We also note that automatically labeling features such as factuality and relevance at scale is challenging (for example, how should the factuality of a source be defined?) and therefore deserves a dedicated, in-depth study of its own, which we leave to future work. The study suggested by the reviewer is a promising future work and we revised the Limitations and Future Work section accordingly (Lines 492-510).
>
> ---
>
> **[Q1] Reasoning Model Behavior**
>
> We appreciate the reviewer’s comment on Figure 3. Our analysis of reasoning traces is currently qualitative: we manually inspected a set of examples and the traces are generally consistent with the cited sources. We agree that automatically detecting errors in the reasoning trace, identifying the “divergence” points, and potentially backtracking is an exciting study of reasoning models under search / RAG settings. We leave the above analysis to future work, as stated in Section 3 (Lines 242-243).

---

### Official Review · Reviewer_7PyT · 2025-11-01

**Soundness:** 3
**Presentation:** 3
**Contribution:** 3
**Rating:** 6
**Confidence:** 4

**Summary:**

This study aims to solve the issue of evaluation benchmark for search-augmented LLM limited in scale and narrow in scope, often constrained to static, single-turn, fact-checking questions. To this end, a crowd-sourced, large-scale, human-preference dataset is proposed together with the analysis and performance accessing across different categories and settings. Overall, this is an important contribution in evaluating retrieval-augmented generation.

**Strengths:**

1. Unlike prior datasets such as SimpleQA and BrowseComp which are static, English-only, single-turn fact-seeking queries, the proposed Search Arena evaluates models in diverse, open-ended, multilingual, and multi-turn settings.

2. The human-preference analysis are detailed, covering number of citations, supportive claims, cited sources, etc. This is the crucial point in terms of the construction principle of Search Arena.

3. A set of detailed experimental results and analysis are provided to cover various aspects.

**Weaknesses:**

1. The reliability of the collected data should be further judged, although the more sophisticated approaches are expensive.

2. The definition of search-augmented LLM and retrieval-augmented generation should be further distinguished if the author discusses them in different context as shown in the related work.  Besides, the conversational search [1,2] is highly related to the proposed Search Arena, i.e., multi-turn human-AI interaction in search setting.

3. Section 2 discuss the difference compared to SimpleQA, BrowseComp, and Text Arena, but lack of the comparison with existing conversational search/RAG datasets, e.g., TopiOCQA [3], Coral [4].

[1] Neural Approaches to Conversational Information Retrieval. SIGIR 2020.

[2] A Survey of Conversational Search. ACM TOIS 2025.

[3] TopiOCQA: Open-domain Conversational Question Answering with Topic Switching. TACL 2021.

[4] CORAL: Benchmarking Multi-turn Conversational Retrieval-Augmentation Generation. NAACL 2024.

**Questions:**

1. What is the potential mechanism to improve the reliability of the constructed datasets based on human votes?

2. What is the difference between search-augmented LLM and retrieval-augmented generation (in conversational scenarios)?

---

> ### Author Response · Authors · 2025-11-22
>
> Thanks for recognizing the contributions of our work, including the utility of the dataset and the detailed analysis of human preferences. We would like to address your concerns below:
>
> ---
>
> **[W1, Q1] Reliability of the Collected Data**
>
> We would first like to restate that the central goals of the paper are (1) collecting a diverse dataset of human interactions with search-augmented LLMs and (2) conducting a thorough analysis on how users interact with these models and how different features of the models’ responses correlate with users’ preferences.
>
> We would like to clarify that we do not treat the human-preferred response as the objectively “correct” one, but rather as the response better aligned with the user’s personal preferences. In our evaluation and data collection setup, the prompt is tied to the judge (i.e., user): this pairing is critical, as only the original user knows the hidden intent behind their prompt. That being said, to show that users do not vote completely randomly, we ran a small study (100 samples with 3 expert annotators) and observed that the agreement rate between expert annotators and users is above chance: 68% when excluding tie votes (random agreement=50%). Since user preferences are subjective and many of the user prompts do not have a clear “correct” answer, we expect to have variability and non-perfect agreement rates. We added the results of this validation study to the Section 3 of the revised submission (Lines 205-212).
>
> ---
>
> **[W2, Q2] RAG vs Search-augmented LLMs**
>
> We view search-augmented LLMs as an extension of traditional RAG systems with the following key differences:
>
> - Compared to traditional RAG operating on a set of given documents, search-augmented LLMs search through the full open web. The search is “unbounded”: there are no ground truth correct set of documents to be retrieved for a given user query due to the dynamic state of the web.
> - Most modern search-augmented LLM systems are implemented as multi-round agent workflows, where the model does multiple rounds of query refinement, search, and generation.
> - In search-augmented chat settings, web search is implemented as a tool. Depending on the input query, the model decides whether to invoke the tool or not (for some prompts, web search might not be required/triggered). Additionally, in our setup, the quality of the final response is impacted by the search module, as well as the tool-calling/agentic and conversational capabilities of the backbone model.
>
> We make the connection to RAG systems more clear in the revised Related Work section (Lines 469-471).
>
> ---
>
> **[W2] Prior Work on Conversational Search**
>
> We agree that conversational search is highly relevant to our current study. We added the suggested prior literature [1, 2] in the revised Related Work section (Lines 467-469).
>
> ---
>
> **[W3] Comparison to Conversational Search Datasets**
>
> Thanks for the suggestion. We would first like to clarify that our setup is different from the one in the suggested conversational search benchmarks, as the latter assume access to a constrained set of documents with explicitly labeled “golden” documents. That said, to broaden the scope of our study, we added comparative analysis between the Search Arena dataset and the CORAL benchmark [4]. We included the results in Appendix G.1 (Lines 1910-1928, Figure G.1) of the revised submission. We also added the suggested prior works on conversational search datasets [3, 4] in the revised Related Work (Lines 467-469).

---

### Author Response · Authors · 2025-11-27

We thank the reviewers for the insightful questions and feedback. We posted our full rebuttal and revised the submission PDF (updates highlighted in red) on Nov 21st.

We appreciate the discussion phase and are happy to address any further questions. Thanks!

---

### Author Response · Authors · 2025-12-03
**Summary of Rebuttal and Revisions**

Dear AC,

We sincerely thank you for managing the review process. We have engaged with the reviewers’ feedback and have updated our submission accordingly on November 21st. For your convenience, we summarize the reviews and our rebuttal below:

---

**Consensus on Strengths**

Overall, all reviewers show positive attitude to our work. They consistently acknowledge the contributions of our work, specifically highlighting the **utility of the large-scale dataset** (7PyT, ttYs, Uoa4, Tf64) and the **depth of the interaction and preference analysis** (7PyT, ttYs, Uoa4, Tf64). They agree that our study addresses a clear gap and offers novel insights into human interactions with search-augmented LLMs.

---

**Addressed Concerns and Revisions**

Reviewers raised four primary areas for clarification, which we addressed as follows:

1. **Reliability of Preference Data (7PyT, ttYs):**
    - *Concern:* Reviewers asked for clarification on the objectivity of user preferences and the reliability of the data.
    - *Action:* We clarified that preferences represent alignment with user intent and preference rather than objective "truth." Crucially, we conducted and added a human validation study (Section 3), demonstrating a 68% agreement rate between expert annotators and users, showing that while human preferences are subjective, the collected votes are not random.

2. **Comparison to Related Benchmarks (7PyT, Uoa4):**
    - *Concern:* Reviewers suggested broader comparisons with related benchmarks and datasets.
    - *Action:* We expanded our Related Work section and added quantitative comparative analyses between our dataset and both CORAL (Appendix G.1) and WildChat (Appendix G.2) to better contextualize our contribution.

3. **Methodological Details & Extensions (ttYs, Uoa4):**
    - *Concern:* Reviewers requested details on the citation attribution pipeline and reasoning model behaviors.
    - *Action:* We provided detailed validation metrics for our LLM-based citation attribution annotation pipeline (83.64% human agreement). We also added a new analysis and findings on the cited domain distribution of reasoning models (Appendix E.2).

4. **Clarification of Objectives, Findings & Future Work (Uoa4, Tf64):**
    - *Concern:* Reviewers sought clarity on the paper’s scope regarding model benchmarking versus interaction analysis, and requested deeper discussion on findings and future directions.
    - *Action:* We explicitly clarified that our primary contribution is analyzing human-model interactions in search-augmented chat settings and identifying preference drivers rather than creating a leaderboard-style benchmark. We also expanded the discussion on specific findings (e.g., negative correlation with Wikipedia citations) and revised the Future Work section to outline how our dataset supports future research, such as augmenting with objective metrics and developing offline evaluations.

---

We believe these revisions have significantly strengthened the submission and thank the reviewers again for their constructive feedback.

Thanks in advance for your time.

Best regards,

The Authors

---

### Meta-Review · Area_Chair_BEB2 · 2026-01-08

**Summary:**

The paper presents "Search Arena," a large-scale dataset of around 24k multi-turn interactions with search-augmented LLMs, including human preference votes. The authors use this data to analyze user behaviors, revealing that users often prefer responses with more citations (even if irrelevant) and tend to dislike Wikipedia sources. They also conduct cross-arena experiments showing search-augmented models perform well in general chat, while standard models struggle in search settings.

**Reviewer Concerns:**

Addressed:

Comparison to benchmarks (7PyT, Uoa4): Reviewers asked for comparisons to CORAL and WildChat. The authors added these quantitative analyses in the appendix, distinguishing their "unbounded search" setup from document-constrained RAG.

Reliability of Preference Data/Annotation (7PyT, Uoa4): Concerns were raised about the subjectivity of user votes and the validity of the LLM-based citation attribution. The authors added a human validation study (68% agreement with users) and validated the citation pipeline (83% agreement), which clarifies the data quality.

Ethics/Privacy (Tf64): Reviewer flagged privacy concerns regarding PII. The authors clarified the filtering process and the takedown mechanism. The Ethics AC also reviewed this and found no issues.

Outstanding:

Objective Correctness vs. Preference (ttYs, Tf64): While the authors acknowledge this, the dataset remains fundamentally a "preference" dataset, not a "factuality" benchmark. The issue that users can be "tricked" by citation quantity remains a feature of the data, not a solved problem.

Lack of Training (Tf64): The paper is purely analytical and creates a resource; it does not demonstrate using the data to train a better model. This remains a limitation in scope.

**Reviewer Scores:**

Reviewer 7PyT (Score: 6): Likely to stay 6 but closer to 8. The authors directly addressed the main weakness regarding the lack of comparison with conversational search datasets like CORAL.

Reviewer ttYs (Score: 6): Likely to stay 6. The concern about human preference conflating credibility with correctness is valid. The authors explained this is a finding rather than a bug, but it limits the utility for evaluating "true" performance.

Reviewer Uoa4 (Score: 6): Likely to remain the same but closer to 8. The request for validating the LLM pipeline was met with concrete agreement numbers, strengthening the paper's claims.

Reviewer Tf64 (Score: 6): Likely to stay 6. Ethics concerns were cleared, but the reviewer's desire for training experiments and objective metrics wasn't fully satisfied, as the authors deemed it out of scope.

---

### Decision · Program_Chairs · 2026-01-26

Accept (Poster)